



# A new look at the jet-storm track relationship in the North Pacific and North Atlantic

Nora Zilibotti[1], Heini Wernli[1], and Sebastian Schemm[2]

[1]Institute for Atmospheric and Climate Science, ETH Zurich, Zurich, Switzerland
[2]Department of Applied Mathematics and Theoretical Physics, University of Cambridge, Cambridge, UK

**Correspondence:** nora.zilibotti@env.ethz.ch

**Abstract.** The western ocean boundaries of the North Pacific (NP) and North Atlantic (NA) set favourable conditions for upper-level jets and baroclinic weather systems that propagate downstream and form the storm tracks. Despite these similarities between the two ocean basins, distinct forcing mechanisms during the winter season give rise to differences in the jet intensity, structure, and variability, as well as in the storm track activity. In particular, the phenomenon of the NP midwinter suppression

of the monthly averaged storm track activity sparked ongoing discussions about fundamental differences between jet-storm track interactions in the NP and NA. This study introduces an alternative method, which avoids monthly averaging, to study the relationship between the background jet core strength ($U$) and, as a measure of storm track activity, the eddy kinetic energy (EKE), both evaluated in the upper troposphere. With our approach, we find that the $U$-EKE relationship is remarkably consistent across the NP and NA, with previously observed differences largely attributable to the differing timescales of jet

variability in the two basins. For our interpretation, the separate consideration of two distinct timescales is important: On seasonal timescales, baroclinic instability results in an increase of EKE with increasing $U$ from summer to winter. In contrast, on sub-monthly timescales, particularly during winter, EKE decreases with increasing $U$, reflecting the effect of baroclinic conversion. Periods of enhanced baroclinic conversion lead to reduced baroclinicity (quantified by $U$) and high EKE, whereas periods of low baroclinic conversion are followed by high $U$ and low EKE. In this framework, the NP midwinter suppression

of monthly averaged EKE reflects that, in midwinter, $U$ remains persistently high in the NP (because baroclinic conversion is suppressed) while EKE is reduced. In other words, in the NP, jet strength varies predominantly from month to month, whereas in the NA, it varies more within individual months such that the midwinter suppression of the monthly averaged storm track activity is less obvious in the NA. The observed $U$-EKE relationship implies that the jet core strength $U$ alone cannot explain the EKE variability across seasons, and we reveal the additional importance of the jet width, which affects eddy

characteristics. A reduced jet width likely plays a role in deforming and meridionally confining eddies, thereby reducing their baroclinic growth. For jets with comparable weak to moderate core strengths in summer and winter, the summertime jets tend to be narrower, and therefore EKE smaller. Similarly, very strong jets in winter are particularly narrow, which implies reduced EKE, supporting the observed $U$-EKE relationship in winter. Finally, cyclone composites show that the reduced EKE during strong jet episodes in winter is manifested by a reduction in the amplitude of the cyclones' surface pressure anomalies, and, in

particular of their associated troughs and ridges. Therefore, the reduction of EKE with increasing $U$ is not related to a decrease in cyclone frequency, but rather to a reduction in cyclone intensity and the associated upper-level wave pattern.



## 1   Introduction

Jet streams are narrow bands of strong westerly winds in the upper troposphere that form a central component of the large-scale atmospheric circulation. They are typically located between 20 and 50° latitude in both hemispheres and they are more intense in the winter season (Koch et al., 2006). By steering synoptic weather systems and modulating their intensification, they exert a strong influence on the downstream weather. In the Northern Hemisphere, the main storm track regions are found in the western North Pacific (NP) and North Atlantic (NA). The two regions are in many ways similar, exhibiting warm oceanic boundary currents and land-sea contrasts, which lead to strong meridional temperature gradients. These gradients are climatologically associated with intense upper-tropospheric jets and provide favourable conditions for the development of baroclinic weather systems. Despite broad similarities between the NP and NA storm track regions, differences in ocean heat transport and the presence of stronger tropical heating in the western NP lead to distinct dynamical forcings, resulting in different jet maintenance mechanisms (Hallam et al., 2022). The NA jet is predominantly eddy-driven, maintained through the convergence of momentum fluxes associated with transient eddies in the mid-latitudes (Holton, 1992; Pena-Ortiz et al., 2013). In contrast, the NP jet exists in a merged state near the poleward edge of the Hadley cell, where it is sustained by both eddy momentum fluxes and the advection of planetary angular momentum within the Hadley circulation (Lee and Kim, 2003; Pena-Ortiz et al., 2013), which is itself driven by tropical heating.

The differing driving mechanisms of the NA and NP jets result in notable differences in their variability. In the NA, positive eddy–mean flow feedbacks contribute to latitudinal jet shifts that can persist beyond synoptic timescales (Novak et al., 2015) as a consequence of a preferred type of Rossby wave breaking over prolonged periods. In contrast, the merged NP jet, particularly in its strong phases, acts to suppress such feedbacks by limiting the meridional propagation of Rossby waves out of the jet core, known as the wave guide effect (Hoskins and Karoly, 1981; Nakamura and Sampe, 2002). This anchors the NP jet near the poleward edge of the Hadley cell, favouring short-term pulsing in the jet intensity over more persistent latitudinal displacements (Lee and Kim, 2003; Eichelberger and Hartmann, 2007; Lachmy and Harnik, 2014). Beyond differences in jet maintenance, another distinction between the basins is the influence of downstream development: the NA storm track is often seeded by wave activity propagating from the NP storm track (Hakim, 2003; Drouard et al., 2015), whereas such a direct downstream influence from the NA to the NP is not observed. However, upper-level wave packets propagating across Asia have also been shown to affect the NP storm track (Chang and Yu, 1999; Hoskins and Hodges, 2002; Chang, 2005). In particular, Chang (2005) highlight the relevance of two different seeding branches for cyclogenesis in the NP.

Fundamental differences in jet–storm track interactions between the NA and NP have also been discussed in the context of the midwinter suppression, a phenomenon first identified by Nakamura (1992). Nakamura (1992) studied the relationship between monthly mean upper-level baroclinic wave activity (measured in terms of the high-pass filtered geopotential height) and background zonal jet strength over the western NA and NP. Classical baroclinic instability theory predicts an increase in baroclinic wave activity with stronger background jets (Charney, 1947; Eady, 1949). While this expected relationship holds in the NA, Nakamura (1992) found that in the NP, for monthly mean jet velocities exceeding approximately $45\,\mathrm{m\,s^{-1}}$, which typically occur during midwinter, baroclinic wave activity instead decreases with increasing jet strength.





Various explanations have been proposed to account for the observed differences, ranging from regional processes unique to the NP, such as upstream seeding due to orography (Penny et al., 2010; Park et al., 2010), to more general mechanisms involving baroclinic, barotropic, and diabatic influences (Chang, 2001; Chang and Zurita-Gotor, 2007). Support for a general mechanism independent of zonal asymmetries, such as topography, is provided by Novak et al. (2020), who were able to reproduce the

midwinter suppression of eddy activity in an idealized, zonally symmetric aquaplanet simulation. One suggested factor for the observed differences are the lower monthly mean jet velocities in the NA, which rarely exceed the $45\,\mathrm{m\,s^{-1}}$ threshold (Christoph et al., 1997). Supporting this view, Afargan and Kaspi (2017) identified a suppression of storm track activity in the NA during years characterized by a particularly strong jet. A mechanism directly linking this suppression to jet strength was proposed by Nakamura (1992), who suggested that the enhanced group velocity of eddies in midwinter allows wave packets

to exit the baroclinic zone rapidly, thereby limiting their baroclinic growth. While this mechanism was supported by Chang (2001), their results indicated that the increased propagation speed alone cannot compensate the enhanced eddy growth rate, implying that additional jet characteristics may play a role.

Nakamura and Sampe (2002) proposed that a more subtropical jet structure may inhibit eddy development by trapping upper-tropospheric disturbances away from the lower-level baroclinic zone, thereby weakening the vertical coupling necessary for

baroclinic growth. Such a mechanism is further supported by the findings of Schemm and Rivière (2019), who, using theoretical arguments and reanalysis data, showed that the efficiency of eddies in extracting energy from the background baroclinicity is reduced during midwinter. They attributed this primarily to an unfavourable vertical tilt that inhibits efficient baroclinic energy conversion.

The reduction in baroclinic eddy conversion efficiency has also been linked to the horizontal jet structure. Several ideal-

ized modeling studies have shown that enhanced horizontal shear can reduce baroclinic energy conversion via the so-called barotropic governor effect (James, 1987; Nakamura, 1993; Harnik and Chang, 2004; Deng and Mak, 2005). The underlying mechanism is that a narrow and strongly sheared jet acts as a waveguide, confining eddies meridionally within the jet core and setting the waves' meridional wavelength (Hoskins and Karoly, 1981; Branstator, 1983; Ioannou and Lindzen, 1986). According to James (1987), such meridional confinement limits the spatial extent of baroclinic conversion, while concurrent

distortions in the eddies' vertical structure further inhibit growth by reducing the baroclinic conversion efficiency. Additionally, increased horizontal shear has been proposed to enhance the momentum flux from eddies to the mean flow, which may further dampen eddy amplification by accelerating the background jet and stabilizing the flow (Deng and Mak, 2005). While the barotropic governor effect has been considered on inter-annual timescales, its role in modulating storm track activity in response to seasonal variations in jet strength remains debated. In particular, the relatively small seasonal changes in horizontal

jet structure and a temporal lag between changes in shear and suppressed eddy activity have cast doubt on its relevance on monthly to seasonal timescales (Chang, 2001; Harnik and Chang, 2004; Novak et al., 2020). Chang (2001), for instance, emphasized the importance of changes in eddy structure for the suppression of EKE with jet strength on inter-annual timescales, while attributing the seasonal behavior to diabatic processes.

The midwinter suppression of upper-level eddy activity has implications for the characteristics of surface cyclones as well.

For instance, Schemm and Schneider (2018) found a reduction in cyclone lifetimes in midwinter, as well as changes in deep-





ening and decay rates and mean eddy kinetic energy per cyclone life cycle and related it to a reduced parcel autocorrelation from a Lagrangian viewpoint. They argued that the midwinter suppression does not arise as a consequence of a reduction in total cyclone numbers but rather due to changes in the aforementioned internal storm track characteristics. An extension of this argument was given in Schemm et al. (2021) where it was shown that surface cyclones from different genesis regions exhibit

different responses in their intensity, lifetime and baroclinic conversion during midwinter. In particular, the study found that, in midwinter, surface cyclones with genesis above the Kuroshio current undergo a rapid and significant deepening in the early life cycle phase, followed by an accelerated decay as they rapidly propagate poleward from the primary baroclinic zone.

To summarize, previous research has highlighted pronounced differences in the driving mechanisms of the jets in the NP and NA, as well as in their variability and interactions with the storm tracks, particularly in the context of the NP midwinter

suppression. In this study, we aim to demonstrate that, despite these contrasts, the NP and NA jet–storm track relationships are remarkably similar when differences in the timescales of variability are accounted for. We show that including the effect of the meridional jet structure in addition to the jet core strength allows to better understand the jet-storm track relationship in both the NP and NA and its variability across timescales.

The study is organized as follows: Section 2 presents the data and parts of the methods. In Sect. 3 we introduce and apply

a new averaging method to study the jet-storm track relationship, with a special focus on separating different timescales of variability. Section 4 discusses further jet characteristics that play into the observed relationship, and explains the differences between the seasonal and sub-monthly relationship. Section 5 relates the jet characteristics to eddy and cyclone properties. Finally, our results are summarized and related to findings from past studies in Sect. 6.





## 2 Data and methods

### 2.1 Data

In this study we use ERA5 reanalysis data provided by the European Centre for Medium-Range Weather Forecasts (Hersbach et al., 2020). We consider the period from 1980 to 2022 using data on a 0.5° grid. Variables derived from the wind field are evaluated on pressure levels, while potential vorticity (PV) is computed on isentropic levels. To retain only synoptic- and large-scale features, the wind field is spatially filtered using a spherical harmonics filter with triangular truncation at T80 and Gaussian decay (see Zilibotti (2023); Bukenberger et al. (2025)). This filter attenuates wavelengths shorter than approximately 500 km. To separate different timescales, high- and low-frequency components are computed using a 10-day temporal high- and low-pass filter, respectively. The low-pass filtered zonal wind component is henceforth referred to as the background jet $U$, and the EKE is defined as $\frac{1}{2}(u'^2 + v'^2)$ where $u'$ and $v'$ are the zonal and meridional high-pass filtered wind components, respectively. Figure 1 shows the climatological low-pass filtered zonal wind on 250 hPa and EKE on 300 hPa in DJF.

### 2.2 Choice of domain and variables

At every 6-hourly time step, a representative zonal background jet velocity $U$ is defined separately for the NP and NA storm tracks. To this end, we calculate the zonally averaged latitudinal maximum of the low-pass filtered zonal wind on 250 hPa within the black boxes in Fig. 1 located at (40°W–80°W, 20°N–80°N) for the NA and (140°E–180°E, 20°N–80°N) for the NP. With this measure, we aim to quantify the jet core strength, while the meridional jet structure is accounted for by additionally assessing the effect of jet width in Sect. 4. The jet width is quantified at each longitude of interest as the number of degrees latitude around the jet maximum where the low-pass filtered zonal flow exceeds half the maximum value within 10-70°N. The final jet width is a zonal average over the jet widths at each longitude within each domain. The EKE is averaged on 300 hPa in the red boxes in Fig. 1 for the NP and NA, respectively. The vertical levels for $U$ and EKE are chosen as the pressure levels where the variables are largest in the DJF climatology averaged within the boxes. If we analysed EKE on the same level as the jet, this would yield the same findings. As visible in Fig. 1, the analysis domains in the NP and NA represent the storm track entrance regions and are characterized by strong background jet velocities.

### 2.3 Cyclone perspective

Cyclone statistics and cyclone-centered composites are based on surface cyclone tracks computed with the contour identification and tracking method introduced in Wernli and Schwierz (2006) and refined in Sprenger et al. (2017). We select cyclone tracks that intensify by at least 10 hPa throughout their lifetime, reaching a minimum SLP below 990 hPa and spend a minimum of 24 h in the NP (100°E-120°W, 20°N-80°N) or NA (100°W-0°W, 20°N-80°N), respectively. Finally, from 6-hourly track data, we select the timestep around which the centered 12-hourly SLP intensification is maximal. Cyclones whose center at maximum intensification is within the red NP and NA domains, respectively, are included in the cyclone-centered composites. This allows to select only cyclones that contribute to the EKE field within this domain.



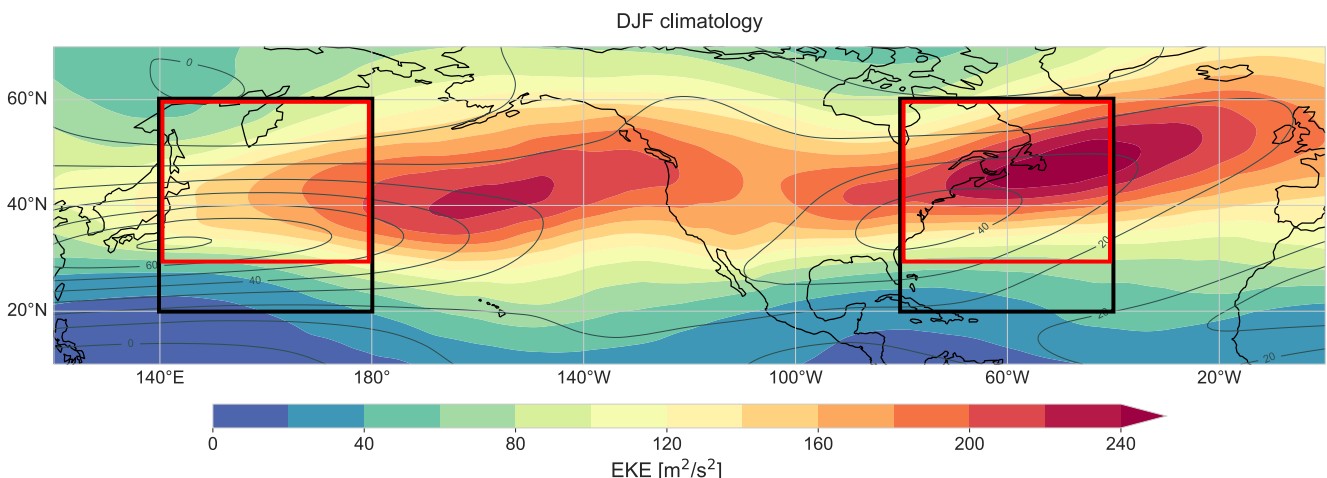

**Figure 1.** DJF climatology of low-pass filtered 250 hPa zonal wind in grey contours (contour interval of $10\,\mathrm{m\,s^{-1}}$) and EKE on 300 hPa in shading (in $\mathrm{m^2\,s^{-2}}$). The black box represents the analysis domain for the background jet and the red box represents the EKE averaging domain.





## 3 The fundamental jet-storm track relationship on two timescales

This section examines the relationship between the jet stream and storm tracks in the NP and NA regions and investigates the mechanisms underlying the differences in this relationship between the two basins reported in previous studies. A new method that accounts for different timescales of variability is introduced to study the relationship between the background jet strength $U$ at 250 hPa and EKE at 300 hPa, which yields a consistent relationship in the NP and NA.

### 3.1 The effect of monthly averaging

We first analyse the relationship between $U$ and EKE using monthly mean values in the western NP and western NA. This approach replicates the analysis presented in Nakamura (1992), where monthly averaged baroclinic wave activity (quantified by the high-pass-filtered upper-level geopotential height) was examined in relation to the monthly averaged background jet strength. The domain and diagnostic measures used to quantify the background jet and baroclinic wave amplitude differ slightly between our study and the one by Nakamura (1992), as described in Sect. 2.2.

Each data point in Fig. 2a,b represents a monthly mean, with monthly means computed around a central date every 10 d. The points are colour-coded according to the month of the central date of the averaging period. The inter-quartile range of EKE for each averaging window of $U$ is shown in light gray. Consistent with the findings of Nakamura (1992), a positive correlation between $U$ and EKE is observed in the NA. In contrast, and also consistent with Nakamura (1992), the NP exhibits an increase in EKE with $U$ up to a threshold jet velocity of approximately $70\,\mathrm{m\,s^{-1}}$, beyond which EKE decreases as jet strength increases. This threshold differs from that reported in Nakamura (1992), due to slightly different definitions of background jet velocity. Notably, mean jet velocities exceeding the threshold of $70\,\mathrm{m\,s^{-1}}$ are rarely observed in the NA. Another important difference between the two basins is that the strongest jet velocities in the NP occur predominantly in January and February, whereas the highest jet speeds in the NA occur throughout the entire extended winter season (see distribution of coloured points in Fig. 2a,b). This suggests that the temporal variability of the jet differs between the NP and NA, which is further explored in the subsequent analysis.

To analyse the variability of jet strength, we classify jet velocities into five equally sized categories. Category 1 (C1) comprises the lowest 20% of background jet velocity time steps during the extended winter season (Nov–Mar), while category 5 (C5) includes the highest 20%. Figure 3 illustrates the contributions of these categories across the extended winter months in the NP and NA. In the NP, the strongest jet categories (C4 and C5) account for approximately 70% of the time steps in January, whereas weaker jet categories dominate in November and March. This indicates a pronounced month-to-month variability in jet strength. By contrast, the NA exhibits a different distribution of jet velocities. In January, the strong jet categories (C4 and C5) make up only about 50% of the time steps, leaving many time steps with weak jet velocities. Apart from November, the category distribution in the NA is rather uniform across months. This suggests that the jet strength in the NA varies more within individual months, whereas in the NP the jet varies predominantly from month to month. The origin of these different timescales of variability will be discussed in Sect. 3.3.





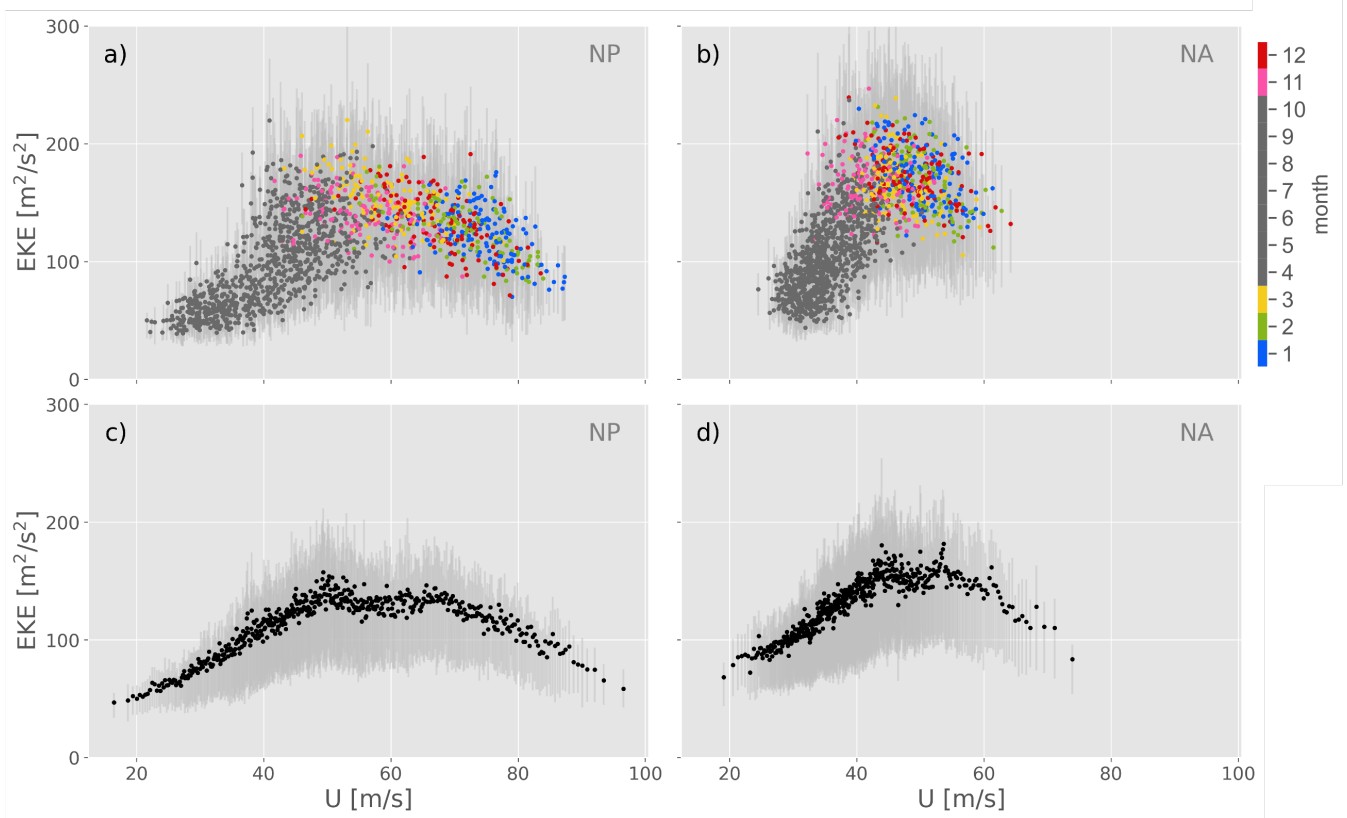

**Figure 2.** Scatter plots of spatial maximum of the 10-day low-pass filtered zonal wind on 250 hPa ($U$) and spatially averaged EKE on 300 hPa in the NP (**a,c**) and NA (**b,d**) for the years 1980–2023. The respective $U$ and EKE domains in the two storm track regions are specified in Sect. 2.2. In (a,b), each data point is an average over 30 d (i.e., 120 consecutive 6-hourly time steps) and coloured by the month of the central date of the averaging window. The 30-d averages are sampled at 10-d intervals. In (c,d), each data point is an average taken over bins of 120 time steps with similar values of $U$. The inter-quartile range of the time steps in each averaging window (a,b) or bin of similar values of $U$ (c,d) is shown in light gray.

The large variability of the NA jet within months has an imprint in Fig. 2a,b, where the effects of monthly averaging obscure the underlying instantaneous relationship between EKE and jet intensity. In the NA, each monthly mean data point represents a wide range of jet velocities, which masks the true signal of how EKE responds to variations in jet strength on short timescales. In contrast, the NP exhibits less variability within the 30-d averaging window, which reduces this masking effect. To tackle the limitations imposed by monthly averaging, an alternative averaging approach is introduced in the following section.

## 3.2 A different averaging method

EKE is, by definition, a highly variable measure, as it quantifies the kinetic energy associated with deviations from the mean flow on timescales up to 10 d. At upper levels, EKE exhibits pronounced fluctuations, peaking with the passage of transient





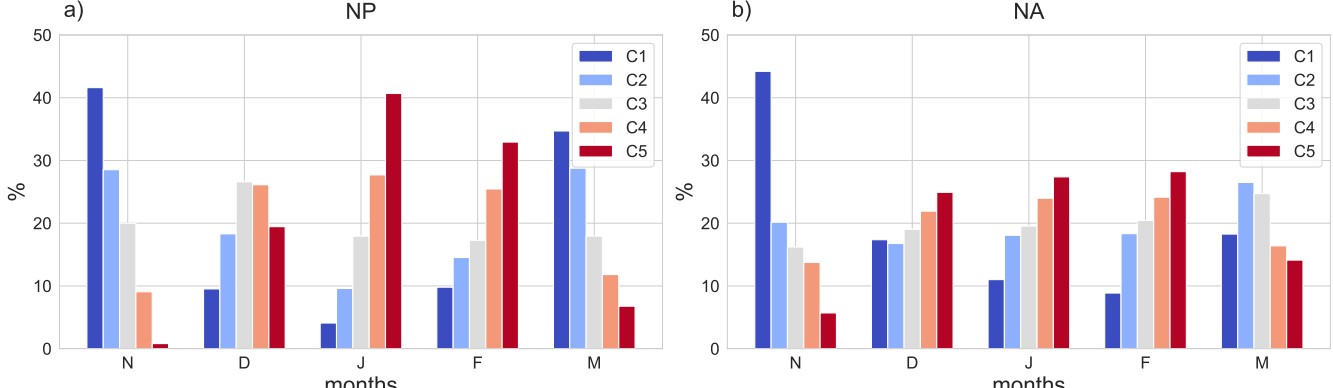

**Figure 3.** Percentage of time steps in each extended winter month that fall into the 5 categories of jet strength (with C1 containing the weakest and C5 the strongest jets) for **(a)** the NP and **(b)** the NA.

troughs and ridges, where the former are often associated with surface cyclones (see for example Fig. 4 in Schemm et al. (2021)). According to baroclinic instability theory, cyclones and their associated upper-level troughs and ridges are expected to intensify most efficiently under conditions of high baroclinicity. In turn, because of the thermal wind relationship, this is related to strong vertical shear in the background jets and typically high wind speed at upper levels. However, even when conditions are favourable for cyclone development, individual time steps may not exhibit cyclone growth nor a pronounced upper-level trough or ridge. Consequently, periods conducive to cyclone growth contain both instances of low EKE when no cyclone is present within the domain and instances of high EKE when a cyclone, along with its associated upper-level troughs and ridges, traverses the region. Due to the substantial variability in EKE, most studies do not directly relate jet strength to EKE at individual time steps but instead apply temporal averaging over weeks or months, as in Nakamura (1992) and Fig. 2a,b. The following analysis maintains the same number of time steps for averaging as in Fig. 2 to address the issue of high variability. However, rather than averaging over consecutive time steps, the data are now binned according to similar jet velocities, analogously to the five jet strength categories introduced earlier. The first bin comprises the 120 weakest jet velocity time steps (equivalent to 30 days of 6-hourly data), while the final bin contains the 120 strongest jet velocity time steps.

Averaging over these bins of similar jet strength yields the results shown in Fig. 2c,d. The non-linear relationship observed in Fig. 2a,b and Nakamura (1992) for the NP is now even clearer in Fig. 2c. While the NA previously exhibited a predominantly positive correlation between jet strength and EKE, the new binning approach reveals a nonlinear relationship similar to the one in the NP, with a distinct negative relationship emerging for averaged jet velocities exceeding $55\,\mathrm{m\,s^{-1}}$. The inter-quartile ranges (light gray shading) show that the variability of EKE within the averaging bins (Fig. 2c,d) is reduced compared to within the 30-d temporal averaging windows (Fig. 2a,b). Nonetheless, the variability within each bin is large, which underlines that the background jet strength alone can only explain part of the overall EKE variability. The remaining variability can partly be attributed to its transient nature, as elaborated above. However, an important portion of the EKE variability for a given background jet strength can be explained by additional jet characteristics, which will be discussed in the following sections.



The alternative averaging method bears multiple advantages. First, by grouping time steps with similar jet velocities, it avoids the issue of signal masking that arises when averaging over a wide range of jet strengths that potentially occur at different days within a month. Second, this approach ensures that rare occurrences of high jet velocities are represented separately, rather

than being masked within averages dominated by time steps with lower values of $U$ flanking transient period of high $U$. As a result, the sparsely distributed data points on the far right of Fig. 2c,d provide a more accurate representation of the jet–storm track relationship at extreme jet strengths.

### 3.3    Two timescales, two relationships

To understand how the non-linear relationship in Fig. 2c,d emerges, the instantaneous $U$-EKE relationship is studied for the
different months separately. The resulting regression lines for January and July are shown in Fig. 4a,b. The correlations in January are $-0.38$ and $-0.23$ for the NP and NA, respectively, and in July $-0.04$ and $-0.17$. To visually represent the $U$-EKE relationship and the explained portion of the variability, the averaging method introduced in Sect. 3.2 is applied to the time steps of the two months separately. For each month, all time steps for the years 1980–2023 are binned according to the background jet strength. The resulting averages and inter-quartile ranges are depicted in addition to the regression lines. While
the averages exhibit a clear negative relationship in January, the inter-quartile-ranges are large, indicating that a large portion of the EKE variability is still not explained by the background jet strength, which explains the low correlations, especially in the NA. Nonetheless, the correlations are highly significant for all months, as determined using a t-test statistic. In Fig. 4c,d the regression lines are overlaid for all months (colour-coded) onto the results previously shown in Fig. 2c,d, allowing for direct comparison.

The results indicate that EKE generally decreases with increasing jet strength. This negative relationship strengthens from summer to winter, as evidenced by the increasing slope magnitude of the regression lines, especially in the NP. Additionally, the entire regression lines corresponding to the individual months shift toward higher $U$ and EKE (the upper-right corner of the diagram) from summer to winter, except for DJF in the NP, where the shift is predominantly toward higher $U$. This behavior can be interpreted as the result of two distinct $U$–EKE processes operating on different timescales. The seasonal shift toward
higher $U$ and EKE can be explained by baroclinic instability theory: as background baroclinicity and jet velocities increase from summer to winter, baroclinic growth is enhanced, leading to stronger storm tracks. This constitutes the first process, occurring on seasonal timescales. The second process occurs on sub-monthly timescales and is responsible for the negative slope observed in the regression lines for individual months.

The co-occurrence of high $U$ with low EKE and conversely low $U$ and high EKE likely reflects the result of baroclinic
conversion. This process transfers energy from the mean available potential energy to the eddy energy, thereby reducing the background baroclinicity (and hence $U$) while promoting eddy growth. As a result, periods of enhanced baroclinic conversion are followed by a weakened $U$ and increased EKE. In contrast, periods of low baroclinic conversion allow for baroclinicity to build up again, leading to high $U$ and low EKE. This behavior is, to some extent, consistent with the nonlinear oscillator model proposed by Ambaum and Novak (2014) for the storm track entry regions. However, this model is highly idealized and
it cannot explain the quasi-persistent state of high $U$ and low EKE observed in the NP during midwinter. This quasi-persistent







**Figure 4.** (**a**,**b**) Regression lines of instantaneous January (blue) and July (red) 10-d low-pass filtered maximum zonal wind at 250 hPa ($U$) and spatially averaged EKE at 300 hPa in the NP and NA domains, respectively. The data points represent averages and inter-quartile ranges of bins with similar values of $U$ as in Fig. 2c,d but for the two months individually. (**c**,**d**) Regression lines as in (a,b) for all months separately superimposed on binned averages from Fig. 2c,d. Shown are results for the NP (a,c) and NA (b,d) for the years 1980-2023. The respective $U$ and EKE domains in the western basins are specified in Sect. 2.2.

state suggests a suppression mechanism by which a strong jet inhibits EKE growth, as widely discussed in the midwinter suppression literature, summarized in the introduction. A key result here is, however, that a negative sub-monthly relationship between $U$ and EKE can be identified with the new averaging method in almost all months and in both storm track regions.

An additional mechanism influencing EKE is also suggested by the seasonal differences in the $U$-EKE relationship: While we expect an overall increase of EKE from summer to winter due to differences in the averaged jet velocities, it is unclear, why for a given background jet velocity (e.g., $40\,\mathrm{m\,s^{-1}}$), the expected EKE is higher in January (given by the y-value of the solid dark blue line at x $= 40\,\mathrm{m\,s^{-1}}$ in Fig. 4c) than in July (given by the solid red line). This suggests variations in jet characteristics





beyond the background jet strength contributing to the seasonal differences in the $U$-EKE relationship. This aspect will be discussed further in Sect. 2.3.

The results presented so far demonstrate that the $U$–EKE relationship is fundamentally similar in the NP and NA. In both basins, the relationship between jet strength and EKE depends on the timescale of analysis. The primary distinction between the NP and NA lies in the timescale of jet variability, which, in turn, leads to the apparent differences in the $U$–EKE relationship observed with monthly averaging. The application of the alternative averaging method, introduced here, substantially reduces these differences and yields a more consistent picture of the underlying processes, as further discussed in the next subsection.

Eventually, the nonlinear relationship observed in Fig. 2c,d and Fig. 4c,d (black dots) emerges as the combined effect of the two processes on different timescales discussed above. Each black dot in Fig. 4c,d represents an average over 120 time steps selected from all months and years between 1980 and 2023, where jet velocities fall within a specific range of $U$ values. These time steps may originate from different years and months. The extent of the monthly regression lines in Fig. 4 indicates which $U$ values are present in which months. For example, in the NP, June, July, August, and September include time steps with $U = 40\,\mathrm{m\,s^{-1}}$.

Consequently, the black dot at $U = 40\,\mathrm{m\,s^{-1}}$ contains time steps from all four of these months. In contrast, the black dot at $U = 70\,\mathrm{m\,s^{-1}}$ primarily consists of time steps from the extended winter months. The increase in EKE between $U = 40\,\mathrm{m\,s^{-1}}$ and $U = 70\,\mathrm{m\,s^{-1}}$ can therefore be attributed to a shift of the contributing months from summer to winter, and therefore the seasonal relationship. In contrast, the black dot at $U = 90\,\mathrm{m\,s^{-1}}$ exclusively represents time steps from DJF.

    Figure 4c,d shows that the decline in EKE between $U = 70\,\mathrm{m\,s^{-1}}$ and $U = 90\,\mathrm{m\,s^{-1}}$ coincides with the regression lines for

DJF, indicating that this decrease is associated with the negative sub-monthly relationship between $U$ and EKE, which is particularly pronounced in winter. Thus, the increase in EKE from small to intermediate $U$ values in Fig. 4 results from the seasonal increase of EKE from summer to winter. In contrast, the decrease from intermediate to high $U$ values arises due to the sub-monthly negative relationship between $U$ and EKE. The threshold at which the sign of the overall $U$–EKE relationship (black dots) changes should not be interpreted as a strict physical boundary at which the relationship fundamentally shifts.

Rather, it marks the transition from seasonal effects dominating at lower $U$ values to sub-monthly effects becoming more influential at higher $U$ values as these typically occur only in a reduced number of months. This interpretation also explains the differences in threshold velocities between the NP and NA.

### 3.4   Comparing the North Pacific and North Atlantic

We now turn to an even more detailed and technically more involved investigation of the jet-storm track relationship in the two

regions. Figure 5 summarizes the similarities and differences between the NP and NA in terms of jet strength (a), jet position (d), EKE (b), and the jet–EKE interaction (c). The variances and covariances of these measures, calculated for the entire year as well as specifically for the extended winter season (NDJFM), can be decomposed—without cross terms (see Appendix A for a derivation of the decomposition)—into two components: (i) the variance of monthly means relative to the total mean (henceforth referred to as variance between months), and (ii) the variance of individual time steps within each month relative to the monthly mean (henceforth referred to as within-month variance). The numbers in the top row of Fig. 5 quantify the

total variances and covariances for both the full year and the extended winter season. Additionally, the second row specifies





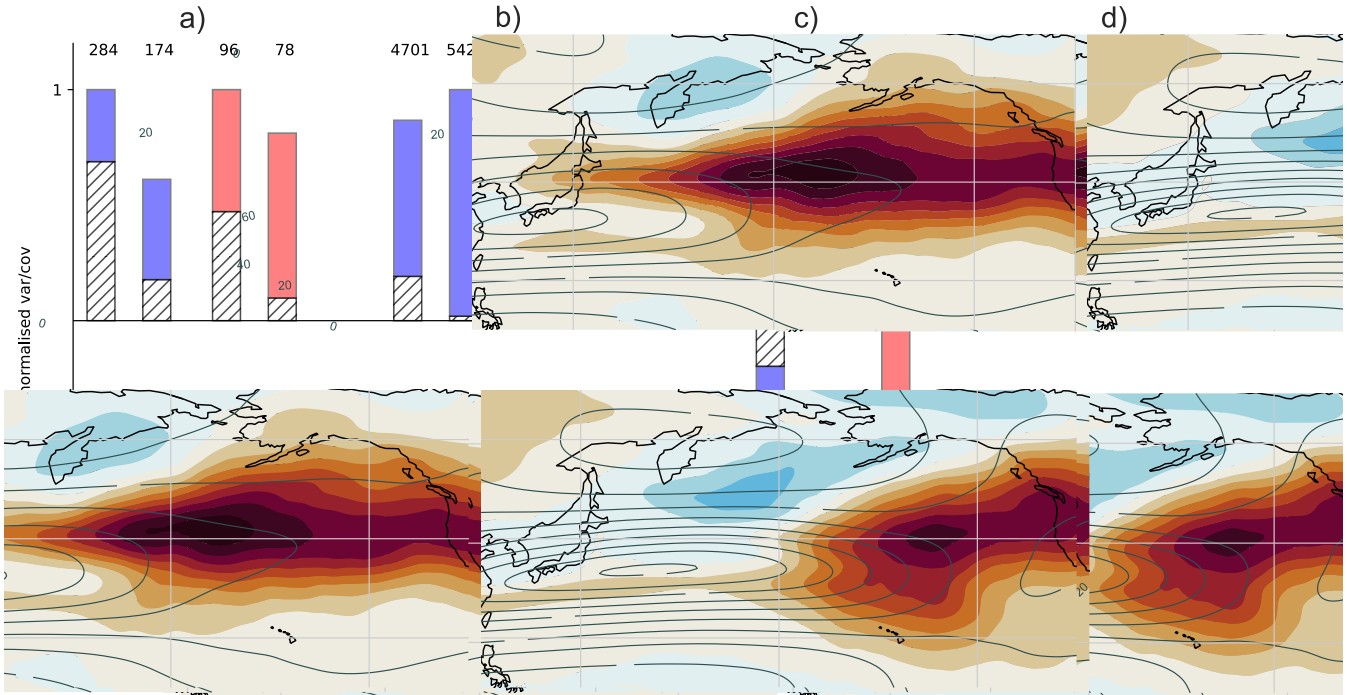

**Figure 5.** Variance of the maximum zonal 10-d low-pass filtered zonal wind at 250 hPa ($U$) and the averaged EKE at 300 hPa, computed over the NP and NA domains as specified in Sect. 2.2. Also shown are the variance of the latitude of the $U$ maximum and the covariance between $U$ and EKE. Variances and covariances are calculated for the entire year and the extended winter months (NDJFM) in the NP (blue) and NA (red). Bars are normalized by the total annual variance of the respective basin. The hatched portion represents the between-month variance and the coloured portion (blue for NP, red for NA) represents the within-month variance. Absolute total variances and covariances are given in the first row, while the second row specifies the correlation associated with the covariances.

the correlations between $U$ and EKE. The bars in the figure are normalized with respect to the total variance/covariance over the entire year for the NP and NA, respectively. The hatched portion of each bar represents the contribution of the variance/covariance between months (term ii/iv in Appendix A), while the coloured portion represents the contribution of the within-month variance/covariance (term i/iii in Appendix A).

The variance of $U$ reveals that the NP jet exhibits greater variability compared to the NA (see numbers at the top of Fig. 5a) throughout the year as well as in winter only, consistent with the higher jet velocities reached in the NP basin. However, while the yearly variance in the NP is predominantly driven by between-month variability, this is not the case for the NA, where the variance is more dominated by within-month variability, in particular during winter (compare the extent of the hatched bars). This finding is consistent with the results shown in Fig. 3. As evident in Fig. 5b, the total variance of EKE is larger in the NA compared to the NP, though the relative contributions from between-month and within-month variance are similar in both basins. Due to the inherently large variability of EKE, the correlation values between $U$ and EKE are relatively small. However, these negative correlations are highly significant, as determined using a t-test statistic. Note that winter EKE is almost





exclusively dominated by the within-month variability for both the NP and NA. Next, the covariance of $U$ and EKE (shown in
Fig. 5b) over the full year exhibits a similar separation into between-month and within-month contributions in the NP and NA.
In both basins, the between-month covariance of $U$ and EKE is strongly positive, reflecting the seasonal relationship governed
by baroclinic instability. This seasonal relationship manifests as the shift toward higher $U$ and EKE from summer to winter,
as seen in Fig. 4. Conversely, the within-month covariance is negative, thereby reducing the overall covariance. This negative
relationship, visible in Fig. 4 as the negative slopes of the monthly regression lines, reflects the sub-monthly interaction between
jet strength and EKE. When considering the full year, the seasonal effect dominates over the sub-monthly effect. Examining
the extended winter months separately reveals stronger differences between the NP and NA. In both basins, the overall $U$–EKE
covariance becomes negative. This occurs because the seasonal effect, which drives a positive covariance, is largely removed
when focusing exclusively on the extended winter months (particularly in the NA). As a result, the negative within-month
effect becomes dominant. This effect is stronger in the NP compared to the NA. In addition, in the NP the between-month
covariance is not entirely removed; instead, it also becomes negative. This phenomenon is what is documented as the midwinter
suppression, which has been observed in the NP but is notably absent in the NA. The NP midwinter suppression reflects the
fact that, in midwinter, $U$ remains high while EKE is reduced throughout entire months compared to the surrounding shoulder
seasons. This reduction occurs because the NP jet remains persistently strong and relatively undisturbed during midwinter, as
evidenced by the lower within-month variability of $U$.

The decrease of EKE with increasing jet strength emerges as a fundamental relationship on sub-monthly timescales in
both the NP and NA and corresponds to the oscillation between periods of high and low background baroclinicity (with $U$
acting as a proxy variable for it), which are characterized by low and high EKE, respectively. Apparent discrepancies in this
relationship can largely be attributed to differences in the dominant timescales of jet variability between the two basins. These
differences, in turn, arise from the distinct nature of the two jets. The eddy-driven NA jet experiences strong eddy-mean flow
feedbacks on the jet latitude and strength (Novak et al., 2015) on sub-monthly timescales. In contrast, the NP jet is more tightly
constrained by the descending branch of the Hadley circulation and is less susceptible to eddy-induced feedbacks (Eichelberger
and Hartmann, 2007). The Hadley cell and its descending branch are strongest and furthest southward in midwinter, leading
to persistently higher $U$ and reduced EKE compared to the shoulder months. This introduces a negative relationship between
$U$ and EKE in the monthly evolution within the winter season that is not found in the NA, where such a relationship is only
present on sub-monthly timescales. These strong contrasts in the jet response are evident in the variance of jet latitude, as can
be seen in Fig. 5, where the wintertime latitude variance within months is shown to be markedly larger in the NA compared
to the NP. The dynamical differences between the two jets likely also contribute to the higher EKE variance and the weaker
negative $U$-EKE correlation within months observed in the NA compared to the NP.




## 4 The effect of jet width

This section examines the additional influence of the meridional jet structure on the diagnosed relationship between $U$ and EKE.
Jet width is found to play a significant role in explaining the seasonal variations in EKE for a given $U$, and also contributes to
the $U$-EKE relationship on sub-seasonal timescales.

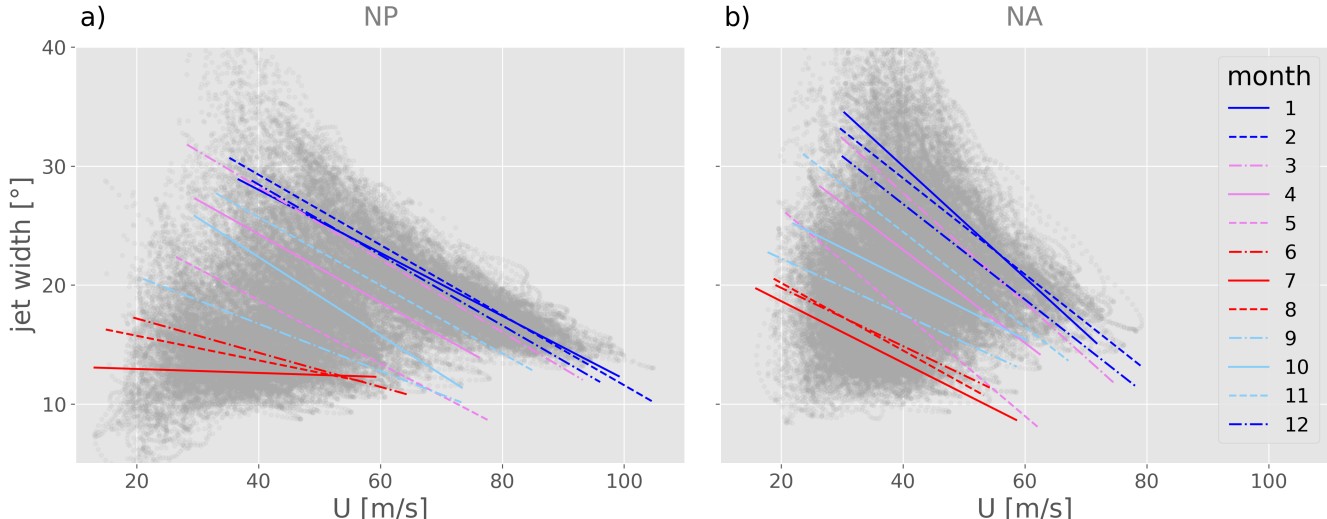

**Figure 6.** The instantaneous relationship between low-pass filtered zonal wind on 250 hPa, $U$, and the jet width shown as a scatterplot (gray
dots) in **(a)** the NP and **(b)** the NA. Regression lines of the data points belonging to the individual months are shown in colour.

To understand how the jet width, quantified by the half-width (see Sect. 2.2), modulates the $U$-EKE relationship, we first ex-
amine the relationship between jet width and $U$ on both seasonal and sub-monthly timescales. Figure 6 shows the instantaneous
relationship between the two variables in light gray, with regression lines for individual months overlaid in colour, separately
for the NP and NA. The distinct positive seasonal relationship, along with the negative sub-monthly relationship identified in
both basins, is strongly reminiscent of the $U$-EKE relationship. Constraints imposed by background forcing through insolation
offer an idealized explanation for the observed relations. For a given month (or season) we may assume a constant differential
heating between equator and pole, leading to a fixed total equator-to-pole temperature difference. Under this constraint, the to-
tal temperature difference can either manifest as a more narrow zone with relatively strong temperature gradients, or a broader
zone with weaker temperature gradients. Following the thermal wind balance, a narrow baroclinic zone with high local temper-
ature gradients translates to a narrow jet with high core velocities $U$. Conversely, a broader baroclinic zone with weaker local
temperature gradients implies a broader and weaker jet. This explains the observed negative sub-monthly relationship between
$U$ and jet width. Now let us choose a fixed wind speed. According to Fig. 6, jets of similar strength are broader in winter.
From summer to winter the equator-to-pole temperature difference is known to increase. If we compare time steps in January
and July that exhibit the same core background jet strength $U$ (e.g., $40\,\mathrm{m\,s^{-1}}$), the thermal wind balance implies that locally





the meridional temperature gradient must be similar. However, as the total equator-to-pole temperature contrast in January is larger compared to July, the region of high temperature gradients must be broader in winter. This explains why for a given $U$, the wintertime jets are broader relative to summer. On average, the increasing equator-to-pole temperature difference from

summer to winter leads to both stronger and broader jets. Appendix B validates this qualitative explanation for the relationship between jet core strength and jet width based on the thermal wind relationship with explicit computations for the NP domain.

Comparing the regression lines in Fig. 2c,d and Fig. 6a,b reveals that the EKE and jet width exhibit a strikingly similar dependence on jet strength on both seasonal and sub-monthly timescales. The observed increase in EKE at fixed jet core strength $U$ from summer to winter can be attributed to the accompanying increase in jet width. As described in Sect. 1, a

narrow jet acts as a waveguide, restricting the meridional extent of eddies and limiting their growth rate. At a fixed core jet strength the more narrow summer jet is thus expected to exhibit lower EKE. The seasonal relationship between $U$ and the jet width can therefore account for a portion of the EKE variability that is not explained by $U$ alone. This effect can be visualized more clearly by extending the analysis presented in Fig. 2c,d by further stratifying the existing $U$ bins—each containing 120 time steps with similar $U$ values—according to jet width. Specifically, each $U$ bin is divided into three equally sized subgroups

based on jet width (narrow, medium, and broad). This stratification allows for the calculation of three separate EKE averages corresponding to different jet widths within each $U$ bin. Figure 7 presents the mean EKE for each $U$ bin (black; as previously) plus for the narrow, medium, and broad jet width categories (red, blue, and pink, respectively), separately for the NP and the NA. The additional explanatory power provided by jet width is immediately evident: For $U < 80\,\mathrm{m\,s^{-1}}$ in the NP and $U < 60\,\mathrm{m\,s^{-1}}$ in the NA, EKE exhibits a strong additional sensitivity to jet width. In particular, the averaged EKE at a fixed

$U$ is clearly below average in the narrow jet category and above average in the broad jet category. In fact, for low to medium values of $U$ the difference between the narrow and broad jet EKE averages is comparable to the inter-quartile range of the $U$-bins depicted in Fig. 2c,d. This implies that a considerable portion of the EKE variability at low and intermediate $U$, which cannot be explained by $U$ alone, is associated with differences in jet width. As discussed in Sect. 3.3, the variability at these $U$ values arises in connection with the seasonal evolution of the $U$–EKE relationship.

At higher values of $U$, corresponding to the descending branch of the $U$–EKE curve in Fig. 7, the stratification by jet width becomes less pronounced. As discussed in Sect. 3.3, the $U$–EKE relationship at these higher velocities is primarily governed by within-month variability during DJF, rather than by seasonality. In DJF, both EKE and jet width decrease with increasing jet strength, with correlations between $U$ and jet width of $-0.82$ in the NP and $-0.63$ in the NA. Note that, especially instantaneously, the jet cannot be regarded as a fixed background state within which the eddies grow; rather, eddies also

significantly shape the jet structure itself.

To explicitly quantify the influence of jet strength and jet width on EKE, we model EKE as a linear function of only $U$ (EKE $= aU + c$), as well as of $U$ and jet width $w$ (EKE $= aU + bw + c$). We do this for 10-d and 30-d averages, in order to focus on the effect of the background jet on the eddies on these two timescales, rather than instantaneous interactions. The explained variance ($R^2$-value) is shown in Tab. 1.

In both ocean basins, the variance explained by $U$ alone is relatively low for both 10-d and 30-d means, with notably lower $R^2$ values in the NP. This reflects the superposition of the two competing effects on different timescales: the seasonal increase




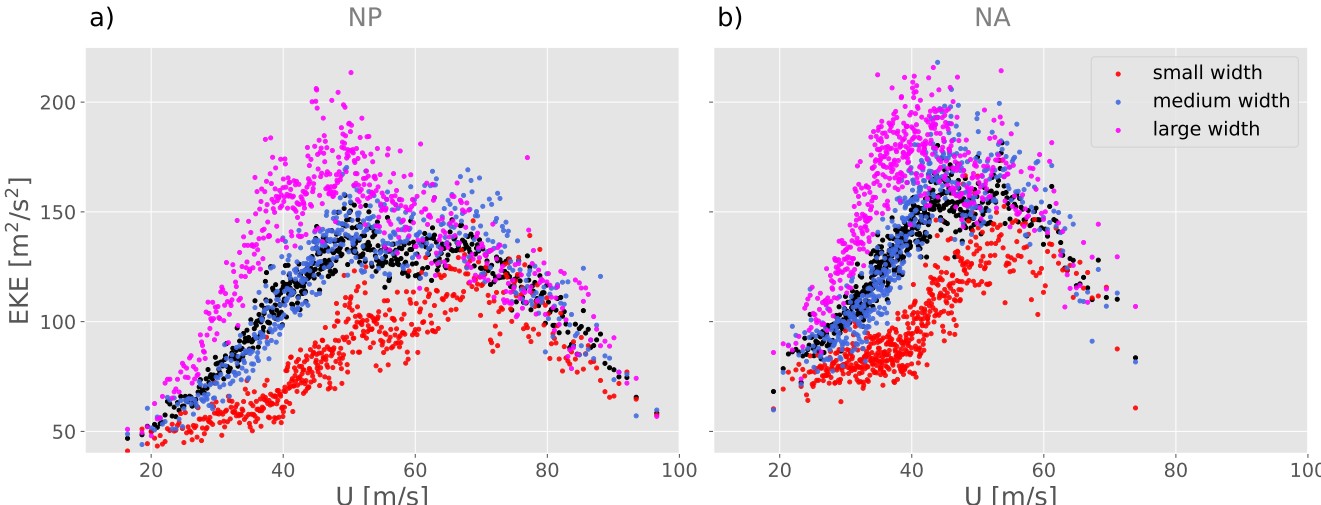

**Figure 7.** Monthly mean values of EKE versus $U$ in **(a)** the NP and **(b)** the NA. Binned averages from Fig. 2 are shown in black. Averages over terciles of different jet width for each $U$-bin are shown in red (narrow jet), blue (medium jet) and purple (broad jet).

| | EKE($U$) (10-day mean) | EKE($U$) (30-day mean) | EKE($U$,$w$) (10-day mean) | EKE($U$,$w$) (30-day mean) |
|---|---|---|---|---|
| **NP** | 0.12 | 0.27 | 0.53 | 0.71 |
| **NA** | 0.21 | 0.45 | 0.52 | 0.73 |

**Table 1.** $R^2$-values of linear regressions (EKE $= aU + c$) and (EKE $= aU + bw + c$) with 10-d and 30-d averaged variables for the NP and NA. The jet width is denoted by $w$ and the jet strength by $U$.

in both $U$ and EKE from summer to winter, and the negative sub-seasonal relationship between the two that is present even after monthly averaging in the NP. These two relationships offset each other and reduce the overall explained variance. In both basins including the effect of jet width strongly increases the explained variance to more than 50% for 10-d averages and more than 70% for 30-d averages. Accounting for the width of the background jet therefore establishes a strong general relationship between the background jet characteristics and EKE. The increase in $R^2$-values with increasing temporal average indicates that combining jet strength and jet width is particularly useful to model the monthly and seasonal evolution of EKE.

The persistent state of a strong and narrow jet associated with low EKE values during midwinter in the NP further suggests that the observed sub-seasonal relationship is not merely a result of an oscillation between short periods of high and low baroclinicity. Rather, it is likely that the NP jet structure during midwinter modulates eddy characteristics in a manner that suppresses their growth, as discussed in previous studies. The following section investigates the properties of eddies and cyclones associated with the two contrasting jet states observed during DJF: the strong and narrow jet characterized by low EKE and the weak and broad jet characterized by high EKE.



# 5  Implications of different jet states in DJF for eddy and cyclone characteristics

To study eddy and cyclone characteristics in the different jet states, we analyse spatial fields of Eulerian eddy measures and
cyclone-centered composites. Rather than focusing on monthly differences, as is common in previous studies on the midwinter
suppression, we retain the framework of binning based on jet strength. Specifically, DJF time steps are stratified into terciles
of $U$, with the analysis focusing on a comparison between the weak and strong jet terciles. The qualitative contrasts between
weak and strong jet conditions are consistent across both the NP and NA basins. Therefore, only composites for the NP domain

are presented here, while corresponding composites for the NA are provided in the Supplement.

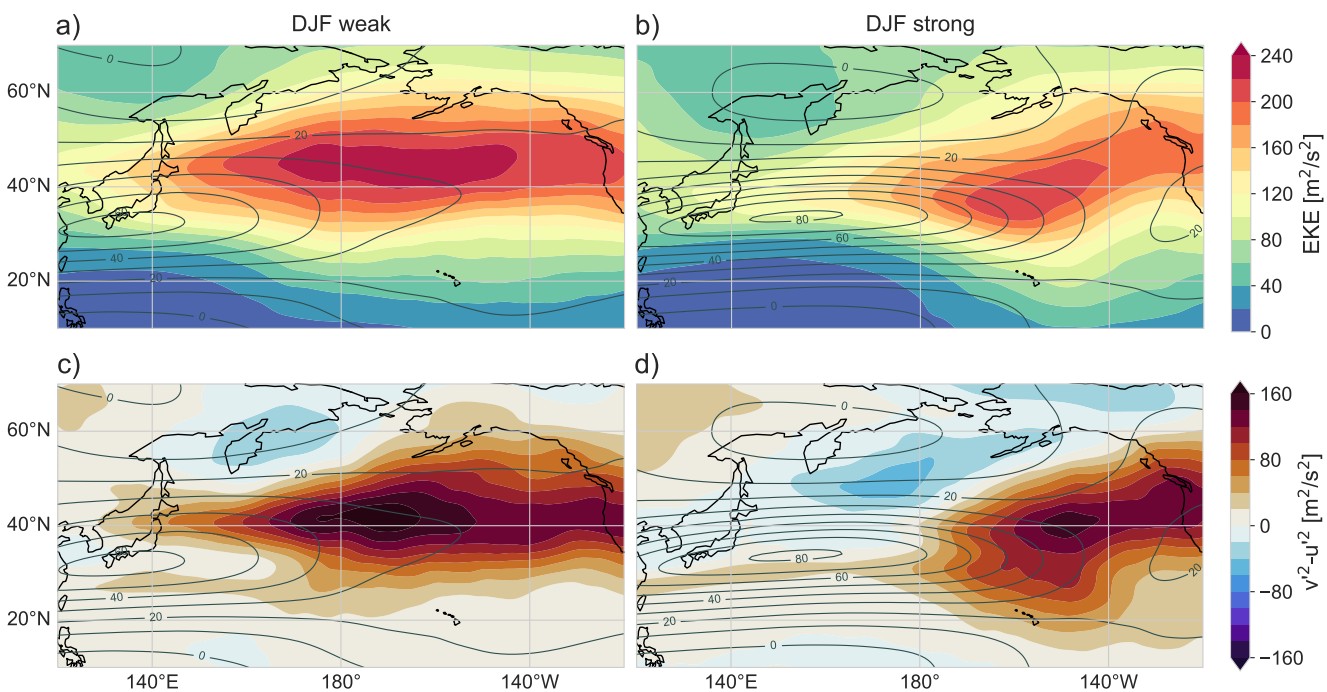

**Figure 8. (a,b)** EKE at 300 hPa (shading) and 10-d low-pass filtered zonal wind at 250 hPa ($U$, gray contours, every $10\,\mathrm{m\,s^{-1}}$) in the NP.
**(c,d)** Eddy orientation measure, $v'^2 - u'^2$ (shading) and the 10-d low-pass filtered zonal wind on 250 hPa ($U$, gray contours). (a,c) are for the
weak jet composite and (b,d) for the strong jet composite.

Figure 8a,b displays the composite mean $U$ and EKE for the weak and strong jet terciles in the NP. In both composites, the
EKE maximum is located in the eastern NP, approximately between 140°W–180°W and 30°N–60°N, and therefore downstream
and slightly north of the jet maximum. A marked reduction in EKE is evident in the strong jet tercile relative to the weak jet
tercile across much of the NP, with the most substantial decrease by approximately one third in the western NP, near and

slightly north-west of the jet maximum (140°E–180°E, 30°N–60°N). This region corresponds to the EKE averaging domain
used in the preceding analyses. A qualitatively similar reduction is observed in the NA (see Supplement Fig. S1). Figure 8c,d
shows the measure $v'^2 - u'^2$ for the weak and strong jet states in addition to $U$. Under certain assumptions (see Hoskins et al.





(1983)) this quantity is approximately proportional to $l^2$-$k^2$, where $l$ and $k$ are the meridional and zonal wavenumbers, making it a measure of eddy anisotropy. Negative values describe zonally elongated eddies, while positive values describe meridionally

elongated eddies. The composites demonstrate that overall the eddies are meridionally elongated in the regions where EKE is high in Fig. 8a,b. The most visible difference between the weak and the strong jet composites is found in the western NP along the northern flank of the jet maximum, which is the region where the EKE decrease is most prominent. In this region the eddies are meridionally elongated on average with weak jets, but, in contrast, slightly zonally elongated with strong jets. Such changes in the eddy anisotropy are known to occur as a consequence of a more sheared and narrow jet. In fact the changes in

eddy anisotropy occur in a region where the horizontal shear appears to be strongly enhanced for the strong jet time steps. This corroborates the influence of the meridional jet structure on eddy characteristics.

Given that EKE is widely employed as a proxy for cyclone activity, we now also establish a connection between the sub-monthly suppression of EKE with jet strength and cyclone characteristics. One way in which cyclones might contribute to the reduction of EKE with jet strength in DJF could be through a reduction of their frequency. To test this, we compare cyclones

in DJF that intensify in the NP and NA regions at time steps that belong to the weak and strong jet terciles, respectively. The time of maximal intensification is defined as the timestep when the centered 12-hourly decrease in minimum sea level pressure (SLP), is largest. The distribution of locations at which cyclones experience maximal intensification reveals that maximal intensification preferentially occurs on the northern jet flank in the western basins, and that the cyclone numbers do not decrease in either basin (but rather increase) with jet strength (see Supplement Fig. S5). Thus, a reduction in cyclone

number cannot account for the observed decrease in EKE under strong jet conditions, confirming the findings from Schemm and Schneider (2018). On the contrary, based solely on the number of cyclones, one might expect an increase in EKE because a large number of cyclones tends to increase the number of days with high EKE. Therefore, the role of surface cyclones in shaping the averaged EKE requires further investigation. To this end, cyclone-centered composites are computed separately for cyclones that intensify in the NP and NA regions during weak and strong jet situations.

The cyclone-centered composites in the NP correspond to the cyclones shown in the two panels of Supplement Fig. S5 within the red box in Fig. 1 at the time of their maximal intensification. The weak jet composite contains a total of 578 cyclones, while the strong jet composite includes 628 cyclones. Figure 9a,b presents EKE, the 10-d high-pass filtered SLP, and the total kinetic energy (KE) at 250 hPa, whereas Fig. 9c,d shows SLP, PV at 320 K, and the high-pass filtered meridional wind at 300 hPa. The significance of the composite-mean PV anomaly, i.e., of the deviation from the mean PV of all DJF cyclones is calculated

using a bootstrapping method, involving 1000 equally sized samples (with a sample size of 578 for the weak and 628 for the strong jet category) of cyclone-centered fields from DJF. Violet stippling indicates grid points with a negative PV anomaly that is significant at the 1% level, while black stippling represents grid points with a positive PV anomaly at the same significance level.

The SLP pattern (shown in Fig. 9c,d) is very similar for both the weak and strong jet cases, with the strong jet cases

exhibiting a slightly lower minimum SLP on average. However, this turns out to be a consequence of the lower background SLP. In fact, the high-pass filtered SLP exhibits a lower minimum value in the weak jet cases. The SLP anomaly with respect to the background is therefore slightly weaker in the strong jet cases ($-12$ hPa in the strong compared to $-16$ hPa in the weak





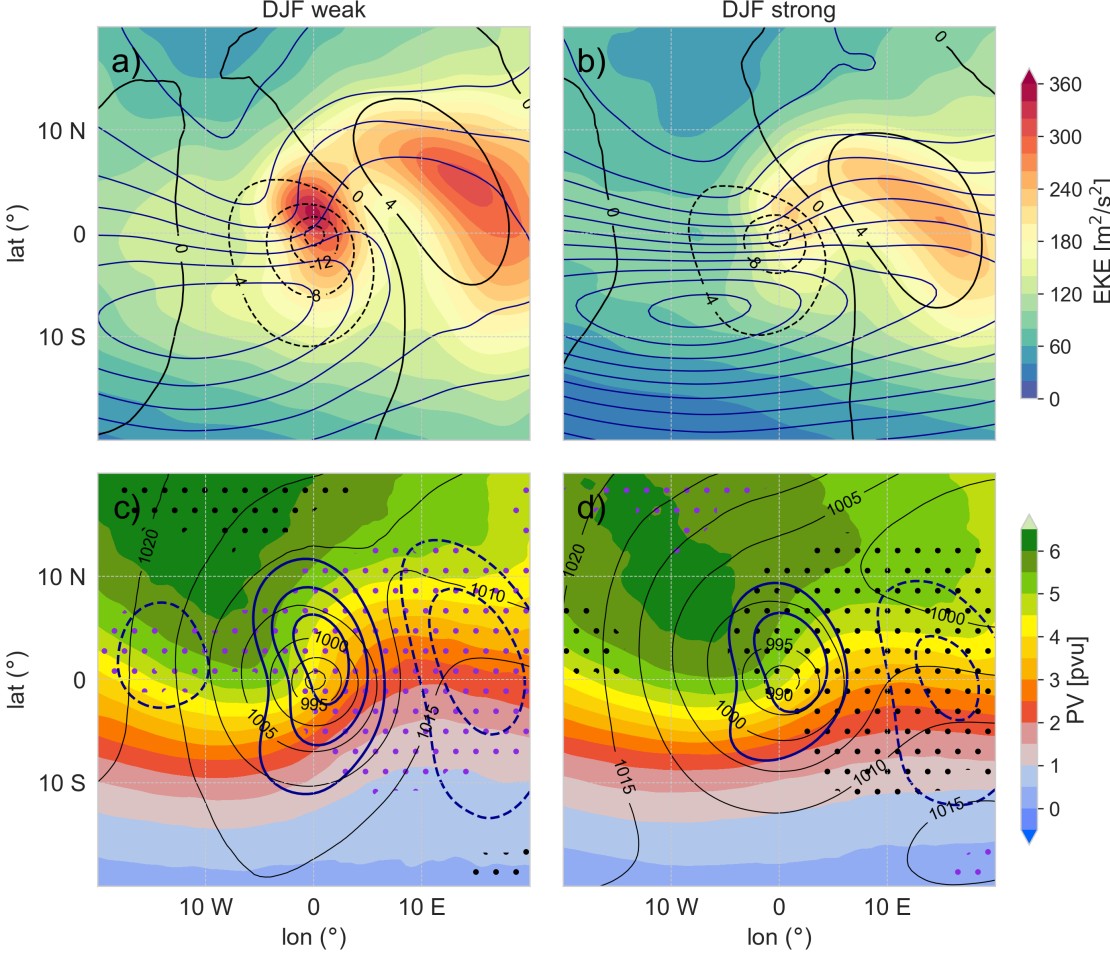

**Figure 9.** Cyclone-centered composites of cyclones at their time step of maximal 12-h intensification in the NP domain. Shown are cyclones with maximal intensification during time steps belonging to the weak (**a,c**) and strong $U$ (**b,d**) terciles. (a,b) show the 300 hPa EKE in shading, the high-pass filtered SLP in black contours (every 4 hPa) and the total kinetic energy at 250 hPa in blue contours (every $200\,\mathrm{m^2\,s^{-2}}$). The innermost total kinetic energy contour is $1400\,\mathrm{m^2\,s^{-2}}$ in (a) and $2200\,\mathrm{m^2\,s^{-2}}$ in (b). (c,d) show PV on 320 K in shading, SLP in black contours (every 5 hPa), and the high-pass filtered meridional wind at 300 hPa in blue contours (every $5\,\mathrm{m\,s^{-1}}$). The significance of a positive (negative) PV anomaly with respect to the PV distribution of all DJF cyclones on a 1% level is shown by black (violet) stippling.

jet cases). The upper-level EKE per cyclone is strongly reduced for the strong jet cases. Thus, the observed decrease in EKE with jet strength cannot be attributed to a reduction in the number of the cyclones, but rather to a reduction of the amplitude of

the upper-level EKE signature of the cyclones (in line with findings from Schemm and Schneider (2018) that the mean EKE per cyclone decreases during the midwinter suppression), which goes along with a reduction of the SLP anomaly. The position of the KE maximum with respect to the cyclone center indicates that intensifying cyclones are preferentially located near the left jet exit region. In the strong jet cases, KE is higher by definition, and its orientation is predominantly zonal, suggesting a





relatively straight jet structure. In contrast, the weak jet cases exhibit a more wavy KE pattern, indicative of a more pronounced
upstream trough and a downstream ridge. This trough-ridge structure is further corroborated by the upper-level PV fields, which
highlight the difference in waviness between the weak and strong jet composites. The statistical significance stippling confirms
that, relative to the DJF climatology, the ridge is more pronounced in the weak jet composite (characterized by anomalously
low PV), whereas it is less pronounced in the strong jet case (characterized by anomalously high PV) at a 1% significance
level.

The amplitude of the trough-ridge structure agrees with the EKE signature and its dependence on jet strength, as shown in
Fig. 9. The EKE exhibits two distinct maxima: one located near the cyclone center and another downstream. These maxima
correspond to the flanks of the downstream ridge that accompanies the surface cyclones at upper levels. These regions are
characterized by strong transient meridional flow, resulting in a pronounced meridional wind perturbation ($v'$). Notably, the
positive $v'$ values close to the cyclone center (but also the negative $v'$ values on the downstream flank of the ridge) are reduced
for the strong jet cases, when the jet is more zonal. This reduction in $v'$ values with jet strength and the more zonal upper-level
structure observed in the KE and PV fields agrees with the observation from Fig. 8c,d that the shape of the upper-level eddies
is affected by the jet strength and meridional structure. The reduction in $v'^2$ is accompanied by a decrease in the meridional
extent of the eddies. The mechanistic understanding gained from previous idealized studies (e.g. Harnik and Chang (2004)),
combined with the observed eddy characteristics associated with the strong jet terciles in DJF, suggest that the reduced jet
width plays a role in deforming and meridionally confining eddies, thereby modulating their baroclinic growth rates within the
winter season. This is not reflected in a reduction in cyclone numbers but rather in a reduction in the amplitude of the associated
upper-level troughs and ridges, as well as the surface pressure anomaly.





## 6   Summary and conclusions

This study investigates the interaction between the upper-level jet and storm tracks in the NP and NA by analyzing the rela-
tionship between background jet strength ($U$) and eddy kinetic energy (EKE). The analysis is conducted using 6-hourly ERA5
reanalysis data for the period 1980–2023. Given the substantial variability of EKE, previous studies have often assessed this re-
lationship by averaging $U$ and EKE over extended periods (e.g., 30 d). However, this approach introduces apparent differences
in the $U$-EKE relationship between the NP and NA, which result from the differing timescales of the jet variability in the two
basins. In particular, in the NA, the strong sub-monthly variability of the jet complicates the comparison between the $U$-EKE
relationships in the NP and NA when using monthly averages. To address this, we introduce an alternative averaging method
in which time steps are grouped by similar $U$ values, enabling an evaluation of the $U$–EKE relationship that is independent
of the timescale of variability. In order to better understand the resulting relationship, we study the sub-monthly and seasonal
effects separately. We additionally examine the role of jet width for the $U$-EKE relationship on both timescales by applying
the alternative averaging approach and timescale separation introduced in the first part of the study. To assess the imprint of
changes in the zonal flow on cyclone and eddy characteristics during DJF, we composite time steps corresponding to low and
high jet strength conditions.

Our findings demonstrate that the $U$-EKE relationship is remarkably consistent across the NP and NA. In both basins the
$U$-EKE relationship differs on seasonal and sub-monthly timescales: On seasonal timescales, EKE increases with jet strength,
consistent with baroclinic instability theory, whereas on sub-monthly timescales, especially in winter, EKE decreases with jet
strength consistent with the idea of an oscillation between high baroclinicity and low EKE followed by baroclinic conversion
building up EKE and decreasing baroclinicity. The previously reported differences in midwinter suppression between the two
basins (Nakamura, 1992, see their Fig. 10) can largely be attributed to differing timescales of jet variability between the NP
and NA and their influence on monthly averaged values. These differences in the time scales are rooted in the underlying jet
maintenance mechanisms (Holton, 1992; Pena-Ortiz et al., 2013; Lee and Kim, 2003; Ambaum and Novak, 2014; Hallam
et al., 2022). In the NP, the jet is primarily driven by the Hadley circulation, which is strongest in midwinter, compared to
the shoulder months. The jet thus stays persistently strong throughout entire months in midwinter, with reduced baroclinic
conversion leading to low EKE. Conversely, the eddy-driven NA jet varies more in its strength on sub-monthly time scales in
tandem with the eddies, such that the inverse relationship between $U$ and EKE is not detectable in monthly averages. Afargan
and Kaspi (2017) showed that a midwinter suppression is present in the NA in years when the jet is particularly strong and
shifted equatorward. These are likely years when the NA jet resembles the NP jet, exhibiting less sub-monthly variability. Our
findings suggest that the inverse relationship of EKE with jet strength is always present in the NA, but generally only evident
on sub-monthly timescales. It is worth noting that the negative sub-monthly correlations are relatively small in magnitude due
to the inherently high variability of EKE; nevertheless, these correlations remain highly statistically significant.

Differences between months in the $U$–EKE relationship that cannot be explained by jet strength alone, as well as the negative
sub-monthly correlation between $U$ and EKE, can be even better understood by additionally considering the effect of jet width.
Jet width and the associated horizontal shear have long been recognized as factors modulating the meridional extent and



meridional wavenumber of baroclinic eddies, thereby affecting their growth rates (Hoskins and Karoly, 1981; Branstator, 1983; Ioannou and Lindzen, 1986; James, 1987; Nakamura, 1993; Harnik and Chang, 2004; Deng and Mak, 2005). Consistent with this understanding, our results indicate that the decrease in EKE with increasing jet strength during DJF is accompanied

by a reduction in the meridional scale of the eddies, which become more zonally elongated. This behavior is likely linked to the narrowing of the jet, given the strong negative correlation between jet strength and jet width observed in DJF. Similar to the $U$-EKE relationship, the relationship between jet strength and width exhibits a pronounced seasonal dependence: for a given jet strength, the jet is wider in winter than in summer. On sub-monthly timescales, however, jet width tends to decrease with increasing jet strength. We propose that the seasonally varying radiative forcing constrains the combination of jet strength and

width. The seasonal broadening of the jet from summer to winter helps explain the higher EKE observed at a given $U$ during winter. Conversely, on sub-monthly timescales, the narrowing of the jet with increasing jet strength is associated with low EKE. It is important to emphasize that, particularly on sub-monthly timescales, the 10-d low-pass filtered jet cannot be regarded as a fixed background state within which eddies evolve. While a broader jet may facilitate the development of eddies with a larger meridional extent, these eddies in turn, modify the jet structure (Lorenz and Hartmann, 2001).

Our findings are consistent with previous work by Harnik and Chang (2004), which highlighted the role of jet width in modulating EKE in an idealized setup. However, Harnik and Chang (2004), Chang (2001), and Novak et al. (2020) emphasized this influence primarily on inter-annual timescales, questioning its relevance for the midwinter suppression on seasonal scales. In particular, Novak et al. (2020) argued that a time lag exists between changes in jet shear and the onset of the suppression of eddy activity in midwinter, based on results from idealized simulations for monthly timescales. Harnik and Chang (2004)

argued that the differences in jet width between midwinter and the shoulder months is relatively small compared to inter-annual variations. Our analysis reveals clear differences in eddy and cyclone characteristics depending on the jet structure in DJF, indicating a potential influence of the jet width on sub-seasonal timescales. However, since our composites aggregate all DJF time steps across multiple years, the diagnosed eddy characteristics reflect both sub-seasonal and interannual variability. Moreover, the background jet in our analysis should not be interpreted as a basic state in the sense used in idealized channel

model experiments (e.g., Harnik and Chang, 2004). Nonetheless, our findings underline the tight link between jet width and EKE on multiple timescales. The importance of the combined effect of jet strength and jet width and their link to differential heating offer a potential pathway through which changes in radiative forcing under future climate conditions could modulate storm track activity.

Variations in eddy orientation associated with changes in jet width likely influence baroclinic conversion and eddy growth

efficiency by altering the dominant direction of eddy heat fluxes. This aligns with the midwinter reduction in these quantities observed in the affected region by Schemm and Rivière (2019). However, the exact effect of changes in the meridional eddy structure on the eddy efficiency in this context requires further research. Beyond altering baroclinic growth rates, increased horizontal shear accompanied by enhanced eddy deformation may also contribute to stronger barotropic conversion of EKE to mean flow kinetic energy. This mechanism constitutes an additional sink of EKE, consistent with the findings of Deng and

Mak (2005), though it is likely of secondary importance in the storm track entrance region (Chang, 2001).



The variations in eddy characteristics induced by changes in the jet strength and horizontal profile are reflected in the properties of individual cyclones. An analysis of cyclone statistics during DJF indicates that the observed suppression in EKE for a stronger and more narrow jet is not associated with a decrease in cyclone frequency. Rather, the upper-tropospheric EKE per cyclone is substantially reduced, consistent with the findings of Schemm and Schneider (2018), alongside a slight

weakening of the associated SLP anomalies. In terms of PV, the reduction in upper-level EKE is linked to a decrease in the amplitude of associated PV anomalies, reflecting less pronounced transient troughs and ridges. The implication of the reduction in upper-level waviness on the surface characteristics of cyclones and their life cycles requires further investigation.





**Appendix A: Separation of variance into between-month and within-month variance**

Let $X$ and $Y$ denote atmospheric variables available at 6-hourly resolution over a 43-year period. The variance of $X$ is defined

as in Eq. A1, where the index $i$ iterates over all time steps of every month across all years, and $\overline{X}$ denotes the mean of $X$ over all time steps. To facilitate a decomposition of the variance, we introduce a new index $t_m$, which iterates over all time steps of the month $m$ across all years. Using this notation, the sum in Eq. A1 can be rewritten as shown in Eq. A2. The expression in Eq. A2 can be expanded to obtain Eq. A3, where $\overline{X_m}$ represents the mean of $X$ over all time steps (across all years) within month $m$. Further expansion of Eq. A3 yields Eq. A4, where the second cross-term vanishes, leading to the simplified form in

Eq. A5, where $n_m$ denotes the total number of time steps across all years within the month $m$.

   The first term in Eq. A5 represents the contribution to the total variance from the variability of individual time steps within each month around their respective monthly means (referred to as the within-month variance). The second term captures the contribution from the variability of the monthly means relative to the overall mean (referred to as the between-month variance), weighted by the number of time steps in each month.

An analogous decomposition can be performed for the covariance between $X$ and $Y$, resulting in a separation into within-month and between-month covariance terms, as shown in Eqs. A6–A10.

$$\mathrm{Var}(X) = \frac{1}{N-1}\sum_i^N (X_i - \overline{X})^2 \tag{A1}$$

$$= \frac{1}{N-1}\sum_m \sum_{t_m} (X_{m,t_m} - \overline{X})^2 \tag{A2}$$

$$= \frac{1}{N-1}\sum_m \sum_{t_m} (X_{m,t_m} - \overline{X_m} + \overline{X_m} - \overline{X})^2 \tag{A3}$$


$$= \frac{1}{N-1}\left(\sum_m \sum_{t_m} (X_{m,d} - \overline{X_m})^2 + 2\sum_m (\overline{X_m} - \overline{X})\sum_{t_m}(X_{m,t_m} - \overline{X_m}) + \sum_m \sum_{t_m} (\overline{X_m} - \overline{X})^2\right) \tag{A4}$$

$$= \underbrace{\frac{1}{N-1}\sum_m \sum_{t_m} (X_{m,t_m} - \overline{X_m})^2}_{(i)} + \underbrace{\frac{1}{N-1}\sum_m n_m (\overline{X_m} - \overline{X})^2}_{(ii)} \tag{A5}$$





$$\mathrm{Cov}(X,Y) = \frac{1}{N-1}\sum_i^N (X_i - \overline{X})(Y_i - \overline{Y}) \tag{A6}$$

$$= \frac{1}{N-1}\sum_m \sum_{t_m}(X_{m,t_m} - \overline{X})(Y_{m,t_m} - \overline{Y}) \tag{A7}$$

$$= \frac{1}{N-1}\sum_m \sum_{t_m}(X_{m,t_m} - \overline{X_m} + \overline{X_m} - \overline{X})(Y_{m,t_m} - \overline{Y_m} + \overline{Y_m} - \overline{Y}) \tag{A8}$$

$$= \frac{1}{N-1}\left(\sum_m \sum_{t_m}(X_{m,t_m} - \overline{X_m})(Y_{m,t_m} - \overline{Y_m}) + \sum_m(\overline{Y_m} - \overline{Y})\sum_{t_m}(X_{m,t_m} - \overline{X_m})\right.$$

$$\left. + \sum_m(\overline{X_m} - \overline{X})\sum_{t_m}(Y_{m,t_m} - \overline{Y_m}) + \sum_m \sum_{t_m}(\overline{X_m} - \overline{X})(\overline{Y_m} - \overline{Y})\right) \tag{A9}$$

$$= \underbrace{\frac{1}{N-1}\sum_m \sum_{t_m}(X_{m,t_m} - \overline{X_m})(Y_{m,t_m} - \overline{Y_m})}_{\text{(iii)}} + \underbrace{\frac{1}{N-1}\sum_m n_m(\overline{X_m} - \overline{X})(\overline{Y_m} - \overline{Y})}_{\text{(iv)}} \tag{A10}$$

## Appendix B: Relating the meridionally integrated upper-level zonal jet within a domain to the temperature difference at the domain boundaries

To formalize the argument from Sect. 4 that the total equator-to-pole temperature difference sets a constraint for the combination of jet core strength and meridional jet structure, we apply the thermal wind relationship to the low-pass filtered temperature ($T$) and zonal wind ($U$) fields:

$$\frac{\partial U}{\partial \ln p} = -\frac{R}{f}\frac{\partial T}{\partial y}. \tag{B1}$$

Vertical and meridional integration yields

$$\underbrace{\int_{y_1}^{y_2} fU(p_1,y)\,dy,}_{\text{(i)}} = \int_{y_1}^{y_2} fU(p_0,y)\,dy - R\int_{y_1}^{y_2}\int_{p_0}^{p_1}\frac{1}{p}\frac{\partial T}{\partial y}\,dp\,dy \tag{B2}$$

$$= \int_{y_1}^{y_2} fU(p_0,y)\,dy - R\int_{p_0}^{p_1}\frac{1}{p}\Delta T(p)\,dp \tag{B3}$$

$$\approx \underbrace{-R\int_{p_0}^{p_1}\frac{1}{p}\Delta T(p)\,dp,}_{\text{(ii)}} \tag{B4}$$

where $y$ represents the meridional position, $f$ denotes the latitude-dependent Coriolis parameter, and $R$ is the specific gas constant for dry air. Equation B4 approximates Eq. B3 under the assumption that the integrated low-level jet is much weaker



than the upper-level jet. Under this assumption the meridionally integrated zonal background jet weighted by the Coriolis parameter within a given domain is proportional to the ($p^{-1}$)-weighted vertically accumulated total temperature difference between the domain boundaries beneath the jet. We apply this to the 6-hourly zonally averaged fields within the NP domain (black box in Fig. 1), using 20°–60°N as integration boundaries for (i) (evaluated at 250 hPa) and 900 hPa–250 hPa as integration boundaries for the integral (ii). Figure B1 shows the distribution of values of the integrals (i) (unhatched boxes) and (ii)

(hatched boxes) for each month. As expected, (i) and (ii) evolve together, both exhibiting a distinct seasonal cycle. The slightly higher values of (i) are likely a consequence of neglecting the low-level zonal jet (first term on the left-hand side in Eq. B3). The sub-monthly variance is substantial, underlining that the fixed equator-to-pole temperature difference within months is a highly idealized assumption, especially if applied to the latitudes 20°–60°N. Nonetheless, the seasonal variance dominates over the sub-monthly variability, as evidenced by the comparatively small extent of the boxes representing the inter-quartile

range within months. As a consequence, the meridionally integrated $fU$ at 250 hPa within our domain can be assumed to be approximately fixed within months. This implies that if the zonal jet core velocity is reduced, the jet must become broader. The effect of $f$ is that a more equator-ward jet must be stronger. The higher values of (i) in winter compared to summer imply that a jet with the same core strength in the different seasons must be broader in winter. Thus, this quantitative approach confirms the qualitative explanation for the relationship between jet strength and jet width provided in Sect. 4. Further support is provided

by the fact, that the order of the monthly mean values of (ii) (shown as horizontal coloured lines) captures the order of the shift of the regression lines in Fig. 6 toward higher $U$ and jet widths from summer to winter.



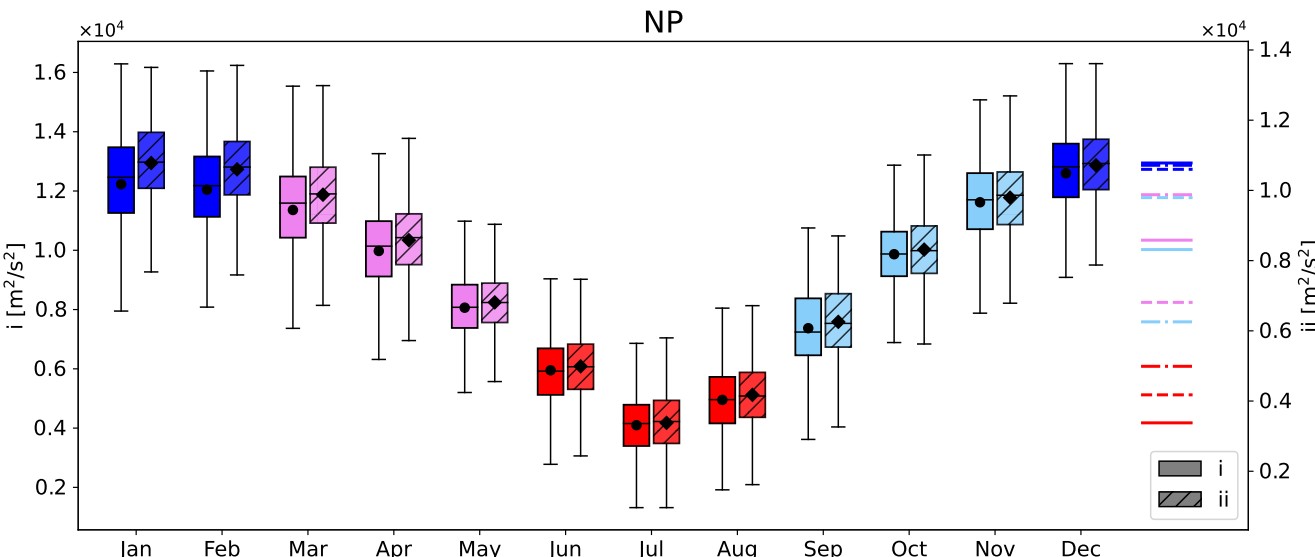

**Figure B1.** Boxplot showing the distribution of expression (i) in Eq. B2 and expression (ii) in Eq. B4 with $y1 = 20°\mathrm{N}$, $y2 = 60°\mathrm{N}$ and $p_0 = 900\,\mathrm{hPa}$, $p1 = 250\,\mathrm{hPa}$. The boxes indicate the interquartile range (IQR), with the horizontal line depicting the median and the black marker showing the mean. Whiskers extend to the most extreme data points within 1.5 times the IQR. The horizontal coloured lines on the right show the averages of expression (ii) for the different months in same colours and linestyles as in Fig. 4.



*Code and data availability.* The ERA5 data are openly available on https://doi.org/10.24381/cds.adbb2d47 (Hersbach et al., 2020). The code used for this study will be made available alongside the manuscript via the ETH research collection after publication.

*Author contributions.* NZ performed all analyses and wrote the paper, based on discussions with and input from all co-authors.

*Competing interests.* At least one of the (co-)authors is a member of the editorial board of *Weather and Climate Dynamics*. The authors declare no other competing interests.

*Acknowledgements.* The authors acknowledge funding from the Weizmann-ETH Zurich Bridge Program 2022 (ETH Zurich Foundation project number 2022-HS-372). The authors are greatful to Michael Sprenger for providing the cyclone tracks and compositing tool. We also thank Gwendal Rivière for fruitful inputs that helped with the mechanistic understanding. A large language model was used for the
improvement of the quality and style of writing.



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
