# Peer review of "A new look at the jet-storm track relationship in the North Pacific and North Atlantic"

_EGUsphere, 2025_

## Referee Comment (RC1)

**"A New Look at the Jet-Storm Track Relationship in the North Pacific and North Atlantic"**

**Authors:** Zilibotti, Wernli, and Schemm

**Recommendation:** Major Revisions

**Overview:**

The authors employ a novel averaging method for the background zonal wind and eddy kinetic energy to examine the subseasonal-to-seasonal variability of the North Pacific and North Atlantic jet streams, and their relationship to variability in storm track activity. The results are rather interesting, and highlight similar dynamical relationships that are at play in both basins, with historical theoretical relationships largely associated with different dominant timescales of variability for each jet. In total, the manuscript is very well-written and I appreciated all of the great insight offered by the authors while explaining their results and their efforts connect results to past literature/classical dynamical relationships. My only substantive comment focuses on the potential for the authors to provide a clearer discussion of their methodological approach and the sensitivity of results to that approach. Given that addressing this major comment may require some additional analysis, I am recommending the manuscript undergo Major Revisions.

**Major Comments:**

**1.** The methodological approach could benefit from the inclusion of additional details that improve the reproducibility of the study. For instance, I believe more detail could be provided regarding how the low- and high-pass filters are applied to the datasets within this study (i.e., could it be possible to add in some mathematical formulations, perhaps?). Additionally, I found myself a bit confused trying to understand how the new averaging approach is applied as a function of zonal wind, $U$. For example, are there certain bin sizes used to assign a timestep to a particular zonal wind speed? How are 120 timesteps selected for each wind speed, and what happens if there are not enough timesteps available for a wind speed? Is anything done to ensure timesteps selected for the new averaging methodology are temporally independent? Finally, is there any sensitivity of the results to the size and position selected spatial domains chosen? In addition to discussing these elements with greater precision in the manuscript, it might also be helpful to include a summary table that contrasts the more traditional averaging methodology with that employed in this study.

**Minor / Specific Comments:**

*1. Introduction*
Section-wide: The authors are commended for producing a very insightful and strong motivation for the forthcoming study!

*2. Data and Methods*
L117: Consider indicating why ERA5 data is used on a coarser 0.5-degree grid rather than its original 0.25-degree grid.

L128–130: How sensitive are subsequent results to the chosen spatial domains? For instance, the NA domain captures much more of the landmass over North America compared to the NP domain – does this potentially affect the results? Furthermore, it appears the NP domain for EKE misses the climatological maximum, whereas the NA domain is favorably located with respect to the EKE maximum. Could the authors comment on the degree to which these domain choices impact the results?

L130: How does this metric work under situations in which there may be multiple jets present at a given longitude?

L139–140: Could the authors provide some more motivation regarding these choices for intensification rate and minimum sea-level pressure? For instance, grounding these values in the cyclone climatologies for each region could be a way to provide objective support for their selection.

*3. The fundamental jet-storm track relationship on two timescales*
L174: The distribution during March also looks less similar to the distributions during DJF for the NA, which might be worth mentioning.

L198–206: I am having a bit of difficulty understanding the methodology for Figs. 2c,d, unfortunately. In particular, it is a bit unclear to me how the 120 timesteps are selected and averaged to produce the plots. For example, are the 120 weakest time steps selected across all years or just within each individual year? Given the varied approaches utilized within the manuscript, it might be beneficial to add either a conceptual diagram or table that summarizes the details of the different methodologies in order to help keep everything straight.

L249: I believe the wrong section is referenced at the end of this line.

L255–264: These explanations are really helpful in deducing how the averaging methodology works, and I think some of these details could be added earlier in section 3 to assist with the initial interpretation of results from Figs. 2 and 4.

L295: I believe this reference should be for panel (c) in Fig. 5.

*4. The effect of jet width*
L367: Are these correlations specifically calculated only for values above a certain wind speed? If so, consider specifying that here.

*5. Implications of different jet states in DJF for eddy and cyclone characteristics*
L427–434: Much of this information, save for the details of the statistical significance test, are provided in the Fig. 9 caption and likely could be omitted for brevity.

L450: The wavier jet structure might also argue for a stronger influence from diabatic processes in the weak jet cases too. Might there be a possibility to highlight to this effect as part of the story? Admittedly, this certainly could be an interesting avenue for future work too.

*Figures and Tables:*

Fig. 2: Could the correlations referenced in the text in L156–157 be calculated and plotted on Figs. 2a,b?

Fig. 5: Is there a reason why the normalized values for EKE in panel (b) do not extend to 1 for the full year, but do for the winter? Discussion of the interpretation of these normalized values being equal to 1 (or –1) might assist a reader. Also, how come the hatching in (c) extends across the colored bars? Could the statistically significant correlations be highlighted with an asterisk?

Fig. 8: These results are very interesting, but would it be possible to produce plots that show the difference between the strong and weak jet terciles and evaluate those differences for statistical significance?

Fig. 9: The difference between the violet and black stippling is a bit difficult to differentiate. Could different colors be used, perhaps?

Fig. B1: It is a bit confusing that different *y*-axes are used for terms (i) and (ii), is there a particular motivation as to why different axes are used?

---

## Author Comment (AC1)

**Final author comments for egusphere-2025-3605**

**A new look at the jet-storm track relationship in the North Pacific and North Atlantic**

by Nora Zilibotti, Heini Wernli, and Sebastian Schemm

We are grateful to both reviewers for their detailed and constructive comments that help us further improve the manuscript. Based on the reviewers' suggestions, we will implement the following main changes:

- We will improve the explanations of the methodology (especially, regarding the averaging methods).
- We will test the sensitivity of our results to the choice of domain.
- We will include further relevant studies in the literature review and in the discussion of our results, and more precisely describe the methods in studies closely related to our work.

This document presents the reviewers' comments in blue and our responses in black.

**Reviewer 1**

The authors employ a novel averaging method for the background zonal wind and eddy kinetic energy to examine the subseasonal-to-seasonal variability of the North Pacific and North Atlantic jet streams, and their relationship to variability in storm track activity. The results are rather interesting and highlight similar dynamical relationships that are at play in both basins, with historical theoretical relationships largely associated with different dominant timescales of variability for each jet. In total, the manuscript is very well-written and I appreciated all of the great insight offered by the authors while explaining their results and their efforts connect results to past literature/classical dynamical relationships. My only substantive comment focuses on the potential for the authors to provide a clearer discussion of their methodological approach and the sensitivity of results to that approach. Given that addressing this major comment may require some additional analysis, I am recommending the manuscript undergo Major Revisions.

We thank the reviewer for the constructive and insightful feedback on the manuscript, and we believe that addressing the comments will improve the manuscript. Below, we provide responses to the individual comments and outline how we intend to improve the manuscript accordingly.

**Major Comments:**

The methodological approach could benefit from the inclusion of additional details that improve the reproducibility of the study. For instance, I believe more detail could be provided regarding how the low- and high-pass filters are applied to the datasets within this study (i.e., could it be possible to add in some mathematical formulations,

perhaps?). Additionally, I found myself a bit confused trying to understand how the new averaging approach is applied as a function of zonal wind, U. For example, are there certain bin sizes used to assign a timestep to a particular zonal wind speed? How are 120 timesteps selected for each wind speed, and what happens if there are not enough timesteps available for a wind speed? Is anything done to ensure timesteps selected for the new averaging methodology are temporally independent? Finally, is there any sensitivity of the results to the size and position selected spatial domains chosen? In addition to discussing these elements with greater precision in the manuscript, it might also be helpful to include a summary table that contrasts the more traditional averaging methodology with that employed in this study.

We thank the reviewer for pointing out the information that is missing to ensure reproducibility and comprehensibility of the study. We will add details regarding the filtering employed in the study. In addition, we will illustrate the method with a schematic and extend the discussion.

Regarding the temporal independence of the timesteps aggregated into one bin in the new averaging methodology. The method does not explicitly ensure temporal independence of timesteps within individual averaging bins. However, it increases the temporal independence within the individual bins, compared to the traditional method of averaging over consecutive timesteps. Given the large number of timesteps aggregated into individual bins and the long time series, we expect that the predominant part of the bins will not be dominated by individual events. Nonetheless, the primary goal of the method is not to address the temporal dependence of the data, but rather of assessing the relation between jet strength and EKE more directly without aggregating many different jet strengths. Similarly to past studies the averaged data is only used to visualise the relationship between U and EKE, while statistical measures such as correlations are calculated on the raw (unaveraged) data.

We will specify in the manuscript how the autocorrelation of the raw data is considered when calculating the effective sample size to compute the significance of the correlations in response to a comment by reviewer 2.

Finally, we will repeat some of the presented analyses with slightly modified spatial domains (eastward shifted and larger domains), to show that the results are not sensitive to the choice of the domain within the limits considered.

**Minor / Specific Comments:**

**1. Introduction**

Section-wide: The authors are commended for producing a very insightful and strong motivation for the forthcoming study!

Thank you, we appreciate the positive feedback!

**2. Data and Methods**

L117: Consider indicating why ERA5 data is used on a coarser 0.5-degree grid rather than its original 0.25-degree grid.

Given that this study focuses on synoptic- and large-scale dynamics, variations on scales smaller than 500 km are not relevant to our analysis. In fact, such scales are filtered out using a spherical harmonics filter that truncates such scales. Therefore, variations that would become visible on a finer grid are filtered out by our method. We will add a short justification to the manuscript.

L128–130: How sensitive are subsequent results to the chosen spatial domains? For instance, the NA domain captures much more of the landmass over North America compared to the NP domain – does this potentially affect the results? Furthermore, it appears the NP domain for EKE misses the climatological maximum, whereas the NA domain is favorably located with respect to the EKE maximum. Could the authors comment on the degree to which these domain choices impact the results?

We thank the reviewer for the valid question. The rationale behind the chosen domains is that we are interested in the direct interaction between the jet and synoptic activity (in particular, the linear stage of the Rossby wave life cycle). The domains are therefore selected such that they include the climatological jet maximum, as well as the storm track entrance region where baroclinic instability is dominant (rather than barotropic wave-mean-flow interaction).

Nevertheless, we will test the sensitivity of the U-EKE relationship by repeating our analyses for two additional domain choices in both basins: (i) domains of the same size as in the manuscript but shifted 10 degrees to the east, as well as (ii) larger domains with the same western bound, extending 20 degrees further eastward. Shifting the NP domain by 10 degrees to the east allows to assess whether the larger fraction of landmass captured in the current NA domain compared to the NP domain affects the results, while increasing the size 20 degrees toward the east allows to include regions of climatologically higher EKE in the NP.

L130: How does this metric work under situations in which there may be multiple jets present at a given longitude?

Being based on the 10-day low-pass filtered zonal wind, our metric filters short-lived jet maxima. In addition, by focusing on the western basins, we avoid capturing the (climatological) African jet. If nonetheless two persistent jets are present at a given longitude within our domain, the stronger one is considered, because we detect the wind maximum at each longitude and then average over all longitudes (within our domain). Thus, the jet strength will be dominated by the jet that extends over most of the domain but two jets at the same longitude are not separated from each other by the method. We would argue that this is a useful measure for the background jet's strength in the domain of interest.

L139–140: Could the authors provide some more motivation regarding these choices for intensification rate and minimum sea-level pressure? For instance, grounding these

values in the cyclone climatologies for each region could be a way to provide objective support for their selection.

We thank the reviewer for this comment. We choose a 990 hPa threshold for the minimal central SLP to select only relatively strong cyclones, similar to e.g. Zhang and Colle (2017). Such a threshold truncates the left-skewed intensity distribution near its peak for both basins (see Lodise et al., 2022, and Neu et al., 2013). To ensure that only cyclones exhibiting a clear intensification phase are included, we set the (relatively weak) criterion that each track has to deepen by at least 10 hPa throughout their entire life cycle. We will add these explanations to the manuscript.

3. The fundamental jet-storm track relationship on two timescales L174: The distribution during March also looks less similar to the distributions during DJF for the NA, which might be worth mentioning.

Thank you for this remark, that is a very good point. We will mention this in the manuscript.

L198–206: I am having a bit of difficulty understanding the methodology for Figs. 2c,d, unfortunately. In particular, it is a bit unclear to me how the 120 timesteps are selected and averaged to produce the plots. For example, are the 120 weakest time steps selected across all years or just within each individual year? Given the varied approaches utilized within the manuscript, it might be beneficial to add either a conceptual diagram or table that summarizes the details of the different methodologies in order to help keep everything straight.

We thank the reviewer for pointing out which aspects of the methods need to be clarified. First, the time steps are sorted using the entire time series of 40 years. Next, the weakest 120 time steps are selected and averaged, followed by averaging across the next 120 time steps in the sorted array. Figure R1 might help to better understand the method. The schematic represents an example time series. T1 to T10 represent the bins used for averaging in Fig. 2a,b, where 30 consecutive days are binned together and averaged every 10 days to produce a sliding mean time series. In contrast, Fig. 2c,d represent averages over bins of similar U containing the same number of time steps as T1-T10, in this figure U1-U4. Here we do not produce sliding means to ensure that the different means are largely independent, resulting in a smaller number of averages (4 compared to 10 in this case). We will add more detail to the explanations in the manuscript and support them with the schematic from Fig. R1.

Figure R1: Schematic illustrating the two averaging methods. The blue dots represent an example time series of background jet strength U shown in Fig. 2a,b.. The black and gray brackets indicate the temporal bins used to compute 30-day (120-timestep) running means. The red brackets indicate bins containing 120 timesteps grouped by similar U values, which are averaged to produce the results shown in Fig. 2c,d.

**L249: I believe the wrong section is referenced at the end of this line.**

We will correct this in the manuscript.

L255–264: These explanations are really helpful in deducing how the averaging methodology works, and I think some of these details could be added earlier in section 3 to assist with the initial interpretation of results from Figs. 2 and 4.

We thank the reviewer for this suggestion on how to make the methodology more understandable to the reader. We will add some of the explanations on how the averages in our new method are calculated earlier in Sect. 3.

**4. The effect of jet width**

L367: Are these correlations specifically calculated only for values above a certain wind speed? If so, consider specifying that here.

There is no threshold used for windspeed, it is simply calculated for DJF.

5. Implications of different jet states in DJF for eddy and cyclone characteristics L427–434: Much of this information, save for the details of the statistical significance test, are provided in the Fig. 9 caption and likely could be omitted for brevity.

Thank you for pointing this out, we will omit parts of the information already provided in the caption.

L450: The wavier jet structure might also argue for a stronger influence from diabatic processes in the weak jet cases too. Might there be a possibility to highlight to this

effect as part of the story? Admittedly, this certainly could be an interesting avenue for future work too.

Thank you for this interesting idea. Indeed, the wavier jet structure is related to more meridional and thus ascending motion, which is likely linked to stronger diabatic processes. These in turn might amplify the downstream ridge. This is particularly interesting when considering the increasing importance of diabatic processes in a warmer climate. We will include this idea in the manuscript.

**Figures and Tables:**

Fig. 2: Could the correlations referenced in the text in L156–157 be calculated and plotted on Figs. 2a,b?

The data points in Fig. 2a,b are 30-day averages computed every 10 days, such that the same timesteps contribute to multiple data points. The points are therefore highly correlated. In addition, the correlations mentioned in the text are dependent on which U-interval is observed. We would thus prefer to keep the statement qualitative. The correlations for 30-day averages without overlap are given in Tab. 1.

Fig. 5: Is there a reason why the normalized values for EKE in panel (b) do not extend to 1 for the full year, but do for the winter? Discussion of the interpretation of these normalized values being equal to 1 (or –1) might assist a reader. Also, how come the hatching in (c) extends across the colored bars? Could the statistically significant correlations be highlighted with an asterisk?

Thank you for pointing this out, we agree that some aspects of this figure require further explanation. Originally the two bars corresponding to one of the basins were normalised with respect to the larger one, but we agree with you that this is confusing. We will adjust the figure such that the bar height is normalised with respect to the all-year bar. Negative values arise because of negative correlations/covariances. In (c) the hatching extends across the coloured bars because the coloured part representing the within-month covariance is negative, thus counteracting the hatched between-months covariance. We will explain these features in more detail in the manuscript. All correlations are statistically significant, but we will specify this in the caption.

Fig. 8: These results are very interesting, but would it be possible to produce plots that show the difference between the strong and weak jet terciles and evaluate those differences for statistical significance?

We will adjust the figure accordingly.

Fig. 9: The difference between the violet and black stippling is a bit difficult to differentiate. Could different colors be used, perhaps?

We thank the reviewer for pointing this out. We will adjust the colour of the stippling.

Fig. B1: It is a bit confusing that different y-axes are used for terms (i) and (ii), is there a particular motivation as to why different axes are used?

The different axes were chosen because neglecting the first term in Eq. B3 leads to differences between the two expressions (i) and (ii) shown in Fig. B1. However, we agree that this can be seen as misleading and will therefore modify the figure to display the same axis for both terms.

**Reviewer 2**

The present paper presents a thorough statistical investigation of the relationship of transient eddy activity (or storm-track activity) with westerly jet intensity and meridional width for the North Atlantic and Pacific. By applying a novel classification method of jet intensity and meridional width to reanalysis data, the authors have revealed that the relationship between the eddy activity and time-mean flow tend to differ fundamentally between inter-monthly and sub-monthly time scales. They have also revealed that for a given jet intensity between weak and modest levels, a narrower jet tends to accompany weaker eddy activity than a broader jet. The findings in this work can be a clue for fully understanding the counterintuitive "midwinter minimum of the North Pacific storm-track activity". I consider this work to be worthy of being published, since the finding of those robust relationships makes an important contribution to the storm-track dynamics, although they still require deeper dynamical understanding. However, I find some aspects of the manuscript that require clarification and improvement as listed below. I therefore consider a minor revision is necessary to properly address those comments before the acceptance of this paper.

We thank the reviewer for the thoughtful and constructive feedback on the manuscript and their overall positive assessment of the work's significance. We greatly appreciate the insight and expertise put into the review. Below, we provide responses to the individual comments and outline how we intend to improve the manuscript accordingly.

**Major scientific comments:**

[A] L100~103: In a line similar to Schemm et al. (2021), Okajima et al. (2022, J. Clim., 35 (4), 1137–1156) investigated the midwinter minimum in the North Pacific storm-track activity. They found that the net eddy conversion/generation rate normalized by the eddy total energy is indeed reduced in midwinter, to which a reduced conversion rate of baroclinic energy gain and an increase rate of barotropic energy loss in midwinter both contribute. It is a relevant work, I believe, to be cited at the end of this paragraph.

We thank the reviewer for pointing us toward this relevant work and will cite it in this context.

[B] L151~166: In these two paragraphs, statistical analyses by Nakamura (1992, N92) are reviewed. For deeper understanding through fairer comparison with this study, the

review should include the following aspects. Specifically, N92 used eddy statistics and mean-flow properties based on sampling not only for individual calendar months but also for 31-day moving windows, which should be mentioned properly, the latter of which is equivalent to the 30-day moving average used in this work. In fact, N92 used the moving average for constructing scatter plots between eddy statistics (activity) and a given mean flow property, including zonal wind speed or meridional temperature gradient. So, any description that may lead to such misunderstanding that N92 only used statistics for individual calendar months should be avoided. Note that Okajima et al. (2022) also used statistics for 31-day moving windows for their analyses. Another important aspect for the scatter plots in N92 is that the mean flow properties were sampled along the instantaneous local storm-track axis defined as latitudinal maxima of eddy amplitude, which differs from the present study. Therefore, the westerly wind speed and associated baroclinicity plotted in the scatter plots in N92 tend to be weaker than the corresponding values in this study especially over the midwinter North Pacific, where the storm-track tends to be located poleward of the intense narrow jet stream.

We will adapt our description of the temporal averaging method in both our study and past work to avoid any misunderstandings. In addition, for a clearer comparison, we will specify the differences between our study and N92 concerning the region where the westerly wind speed is sampled.

[C] L156: "a monthly mean, with monthly means computed around a central date every 10 d" makes no sense, and even "monthly means computed around a central date every 10 d" is quite misleading. This is because "monthly mean" typically signifies averaging over a given calendar month. Going through the manuscript, it will be finally realized that "monthly mean" or "monthly average" in this paper actually denotes "30-day running mean". If this is indeed the case, "a monthly mean, with monthly means computed around a central date every 10 d" should be replaced with "statistics based on 30-d running means evaluated at 10d intervals". In the rest of the manuscript, "monthly mean" should be replaced for clarification with "30-day moving average" or "monthly-scale averaging", wherever necessary. Individual occasions for such replacement are given as "minor comments" below.

We thank the reviewer for pointing this out. Indeed, "30-d running means" is a more precise description of the employed method compared to "monthly means". We will implement these changes in the manuscript.

[D] L236•237: "periods of enhanced baroclinic conversion are followed by a weakened U and increased EKE" sounds quite interesting, but I wonder if such a lead-lag relationship has been robustly extracted in this work through statistical analysis. The same comment applies also to a similar statement in L479~481.

We thank the reviewer for the question. We have performed an analysis examining the lead-lag variation of U and EKE around transitions of U in DJF from its upper to middle tercile. Figure R2 shows the evolution of the 2-day running mean of U and EKE before and after such transitions. This analysis is similar to that of Ambaum and Novak (2014), who investigated the evolution of heat fluxes and baroclinicity and demonstrated that

baroclinicity decreases following spikes in meridional heat fluxes. However, our analysis is centered at different kinds of events, baroclinicity is quantified by U, and instead of heat fluxes we analyse EKE. EKE increases as a consequence of ascent along the sloping isentropes reducing the mean baroclinicity and converting available potential energy to EKE. Consequently, we expect EKE to increase concurrently with a decrease in U. This is what is observed in Fig. R2 and shows that the temporal evolution is in fact as described in the manuscript. We will add this figure to the supplement of the manuscript. We do however not analyse the baroclinic conversion term or APE tendencies explicitly as this is beyond the scope of our work. We therefore formulated the interpretation in the manuscript in a speculative manner.

Figure R2: Evolution of the 2-day running mean of U and EKE after transitions of U from upper to middle tercile in DJF. The solid lines represent the mean of all events at each lag, while the shading represents the inter-quartile range of all events.

[E] L315~318: It is indeed the case that "the NP jet is more tightly constrained by the descending branch of the Hadley circulation" in midwinter. In addition, the NP jet is tightly constrained also by the prominent planetary-wave trough over the Far East, as pointed out by Nakamura et al. (2002, J. Clim.) as well as Nakamura and Sampe (2002, GRL). This aspect should be noted in the text.

We thank the reviewer for this excellent insight. We will discuss the relevance of the planetary-wave trough in the manuscript and cite the mentioned studies.

[F] L339: As stated here, the jet with a comparable strength tends to be broader in winter than in summer (Fig. 6), which is an important finding. From a different viewpoint, the jet with a comparable width tends to be stronger in winter than in summer, reflecting the greater equator-pole temperature difference. This is especially the case over the midwinter North Pacific, where an eddy-driven subpolar jet is merged with the subtropical jet due to the prominent planetary-wave trough over the Far East, as argued by Nakamura et al. (2004, AGU Monogr. 147, 329–345).

Thank you for this comment. We will point out the viewpoint that jets with comparable width tend to be stronger in winter, which is what is frequently observed in the North Pacific during midwinter.

[G] L408: As mentioned here, Fig. 9 clearly shows the tendency for transient eddies over the midwinter North Pacific to be more confined meridionally under the stronger jet than under the weaker jet. This finding is in good agreement with what is indicated in Fig. 3 of Nakamura and Sampe (2002 GRL), which should be cited here. Another important aspect indicated by Fig. 9 is that the strong jet is more zonally extended and cyclone development tends to occur closer to the jet core region if compared to the weak jet condition.

Thank you for pointing this out. We will draw the connection to Nakamura and Sampe (2002 GRL), in the context of the meridional confinement of the eddies. In addition, we will mention both the zonal extension of the jet and the differences in the jet-relative position of the intensifying cyclones.

[H] L485: As commented in [D]. the midwinter North Pacific jet is constrained not only by the Hadley circulation but also by the planetary-wave trough over the Far East. The description here should be modified accordingly.

Thank you for this comment, we will add this aspect to the discussion.

[I] L524•525: The eddy energetics modulated under the intensified narrow jet in the midwinter North Pacific has been investigated also by Okajima et al. (2022), which may be cited here.

We thank the reviewer for pointing this out, and will cite Okajima et al. (2022) in this context.

**Minor comments:**

L22: "narrower with smaller EKE" sounds more appropriate in context than "narrower, and therefore EKE smaller".

We will change the formulation in the abstract accordingly.

L121•122: "10-day temporal high-and low-pass filters" rather than "a 10-day temporal high-and low-pass filter".

Thank you, we will adjust the formulation as suggested.

L167: Better to describe explicitly here if this jet speed classification is based on 10-day lowpass-filtered fields or 30-day running mean fields.

We will clarify this in the manuscript.

p.8, L4 Fig. 2 caption: "30-d moving averages" is better than "30-d averages".

We will implement this change as suggested.

L177: "monthly-scale averaging" is more appropriate than "monthly averaging".

We will change the formulation accordingly.

L181: "simple monthly-scale averaging" sounds more appropriate than "monthly averaging".

We will adjust this as suggested.

L201: It is certainly the case that "a distinct negative relationship emerging for averaged jet velocities exceeding 55 m/s" for the NA in Fig. 2d. This feature is, however, hinted even in Fig. 2b, which should be stated here.

Thank you for pointing this out. We will mention that a negative relationship is hinted also in Fig. 2b.

L222: It should be stated here how the degrees of freedom are estimated for assessing this significance.

Thank you for this interesting remark. We estimated the effective sample size with and without a correction for a lag-1 autocorrelation. The latter was done using formula 2 from Ebisuzaki (1997). Without correction the correlations for all months in both basins are significant on a 1% level. With correction this is the case for all months in the NA, as well as all months except for July and August in the NP, the latter being significant on a 5% level. We will specify the significance levels for the lag-1 corrected effective sample size in the manuscript for clarity, and state that we correct for the lag-1 autocorrelation.

L225: Is this tendency "EKE generally decreases with increasing jet strength" apparent on sub-monthly timescales? If so, it should be stated explicitly here.

The tendency is in fact apparent on sub-monthly timescales, which will be stated explicitly in the manuscript.

L233: "all calendar months" is clearer than "all months". p.11, L3 Fig. 4 caption: "individual calendar months" is better than "individual months".

Thank you, we will implement this change as proposed.

L253: "monthly-scale averaging" is more appropriate than "monthly averaging".

We thank the reviewer for this comment and will adjust the formulation.

L267: "the EKE decrease" is better than "the decrease".

We will implement the change as suggested.

L276~280: Here, the variance/covariance decomposition is described in referring Appendix A. Please clarify that "the variance of monthly means" is based on averages for individual (calendar) months.

We thank the reviewer for this comment and will clarify this in the manuscript.

L351: "narrower summer jet" rather than "more narrow summer jet

We will make this adjustment as suggested.

L358: Based on Fig. 7a, the threshold jet speed over NP for EKE sensitivity to jet width seems to be "70 m/s", rather than "80 m/s" as stated here.

Thank you for noticing. We will change the description of the threshold to 70 m s-1.

L369: "the jet cannot be regarded simply as a fixed background state" sounds more appropriate than "the jet cannot be regarded as a fixed background state".

We will adjust the formulation accordingly.

p.20, Fig. 9: The labels along the abscissa (10E, 10W) and ordinate (10N, 10S) may be misleading. Better to replace them with (+10, -10).

We will replace the labels as proposed.

P.20, L4 Fig. 9 caption: "in thick blue contours" is better than "in blue contours".

We will adjust the caption accordingly.

L441: "a reduction of the SLP anomaly" is ambiguous. "a weakening of the SLP anomaly" is better.

We will change the formulation as suggested.

L442: Isn't it "U maximum" rather than "KE maximum"? "KE" has not been defined yet.

We thank the reviewer for the comment. KE is defined in L428 as total kinetic energy but we will add the abbreviation '(KE)' in the caption of Fig. 9, so that the reader can directly relate it to the figure. We show this quantity in Fig. 9a,b rather than U, because we would like to highlight the instantaneous picture in this discussion.

What is meant by "left jet exit region"? Isn't it simply "jet exit region?

Thank you for this question. We would like to specify that it is the left exit region, as it is known as a region of ascent and cyclone intensification, which is relevant in this context.

Isn't it "U is greater by definition" rather than "KE is higher by definition"? "wavy U pattern" rather than "wavy KE pattern"?

Thank you for this question. We would like to discuss the total wind in this context.

"when the jet is stronger and more zonal" is more specific than "when the jet is more zonal".

We will adjust the text accordingly.

"monthly-scale averages" is more appropriate than "monthly averages".

We will change the formulation as suggested.

L483: "monthly-scale statistics" sounds more appropriate than "monthly averaged values".

We will adjust the formulation accordingly.

**References:**

Ambaum, M.H.P. and Novak, L.: A nonlinear oscillator describing storm track variability. Quart. J. Roy. Meteor. Soc., 140: 2680-2684, https://doi.org/10.1002/qj.2352, 2014.

Ebisuzaki, W.: A method to estimate the statistical significance of a correlation when the data are serially correlated. *J. Climate*, 10, 2147-2153, https://doi.org/10.1175/1520-0442(1997)010<2147:AMTETS>2.0.CO;2, 2017.

Lodise, J., Merrifield, S., Collins, C., Rogowski, P., Behrens, J., & Terrill, E.: Global climatology of extratropical cyclones from a new tracking approach and associated wave heights from satellite radar altimeter. *J. Geophys. Res.: Oceans*, 127, e2022JC018925, https://doi.org/10.1029/2022JC018925, 2022.

Neu, U., Akperov, M. G., Bellenbaum, N., Benestad, R., Blender, R., Caballero, R., Cocozza, A., Dacre, H. F., Feng, Y., Fraedrich, K., Grieger, J., Gulev, S., Hanley, J., Hewson, T., Inatsu, M., Keay, K., Kew, S. F., Kindem, I., Leckebusch, G. C., Liberato, M. L. R., Lionello, P., Mokhov, I. I., Pinto, J. G., Raible, C. C., Reale, M., Rudeva, I., Schuster, M., Simmonds, I., Sinclair, M., Sprenger, M., Tilinina, N. D., Trigo, I. F., Ulbrich, S., Ulbrich, U., Wang, X. L., & Wernli, H.: IMILAST: A community effort to intercompare extratropical cyclone detection and tracking algorithms: *Bull. Amer. Meteor. Soc.*, 94(4), 529-547, https://doi.org/10.1175/BAMS-D-11-00154.1, 2013.

Zhang, Z., and Colle, B. A.: Changes in extratropical cyclone precipitation and associated processes during the twenty-first century over Eastern North America and the Western Atlantic using a cyclone-relative approach. *J. Climate*, 30, 8633–8656, https://doi.org/10.1175/JCLI-D-16-0906.1, 2017.

---

## Author Response (AR1)

**Reply document for egusphere-2025-3605**

**A new look at the jet-storm track relationship in the North Pacific and North Atlantic**

by Nora Zilibotti, Heini Wernli, and Sebastian Schemm

We are grateful to both reviewers for their detailed and constructive comments that helped us further improve the manuscript. Based on the reviewers' suggestions, we implemented the following main changes:

- We improved the explanations of the methodology (especially, regarding the averaging methods).
- We tested the sensitivity of our results to the choice of domain.
- We included further relevant studies in the literature review and in the discussion of our results, and more precisely described the methods in studies closely related to our work.

This document presents the reviewers' comments in blue and our responses and changes in black. The line numbers correspond to the lines in the marked-up manuscript version showing the changes.

**Reviewer 1**

The authors employ a novel averaging method for the background zonal wind and eddy kinetic energy to examine the subseasonal-to-seasonal variability of the North Pacific and North Atlantic jet streams, and their relationship to variability in storm track activity. The results are rather interesting and highlight similar dynamical relationships that are at play in both basins, with historical theoretical relationships largely associated with different dominant timescales of variability for each jet. In total, the manuscript is very well-written and I appreciated all of the great insight offered by the authors while explaining their results and their efforts connect results to past literature/classical dynamical relationships. My only substantive comment focuses on the potential for the authors to provide a clearer discussion of their methodological approach and the sensitivity of results to that approach. Given that addressing this major comment may require some additional analysis, I am recommending the manuscript undergo Major Revisions.

We thank the reviewer for the constructive and insightful feedback on the manuscript, and we believe that addressing the comments helped to improve the manuscript. Below, we provide responses to the individual comments and outline how we improved the manuscript accordingly.

Major Comments:

The methodological approach could benefit from the inclusion of additional details that improve the reproducibility of the study. For instance, I believe more detail could be

provided regarding how the low- and high-pass filters are applied to the datasets within this study (i.e., could it be possible to add in some mathematical formulations, perhaps?). Additionally, I found myself a bit confused trying to understand how the new averaging approach is applied as a function of zonal wind, U. For example, are there certain bin sizes used to assign a timestep to a particular zonal wind speed? How are 120 timesteps selected for each wind speed, and what happens if there are not enough timesteps available for a wind speed? Is anything done to ensure timesteps selected for the new averaging methodology are temporally independent? Finally, is there any sensitivity of the results to the size and position selected spatial domains chosen? In addition to discussing these elements with greater precision in the manuscript, it might also be helpful to include a summary table that contrasts the more traditional averaging methodology with that employed in this study.

We thank the reviewer for pointing out the information that is missing to ensure reproducibility and comprehensibility of the study. We added details regarding the filtering employed in the study. In addition, we illustrated the method with a schematic and extended the discussion (the specific lines in the manuscript are stated under the specific comments).

Regarding the temporal independence of the timesteps aggregated into one bin in the new averaging methodology: The method does not explicitly ensure temporal independence of timesteps within individual averaging bins. However, it increases the temporal independence within the individual bins, compared to the traditional method of averaging over consecutive timesteps. Given the large number of timesteps aggregated into individual bins and the long time series, we expect that the predominant part of the bins will not be dominated by individual events. Nonetheless, the primary goal of the method is not to address the temporal dependence of the data, but rather of assessing the relation between jet strength and EKE more directly without aggregating many different jet strengths. Similarly to past studies the averaged data is only used to visualise the relationship between U and EKE, while statistical measures such as correlations are calculated on the raw (unaveraged) data.

We specified in the manuscript how the autocorrelation of the raw data is considered when calculating the effective sample size to compute the significance of the correlations in response to a comment by reviewer 2.

Finally, we repeated some of the presented analyses with slightly modified spatial domains (eastward shifted and larger domains), to show that the results are not sensitive to the choice of the domain within the limits considered.

Minor / Specific Comments:

1. Introduction
Section-wide: The authors are commended for producing a very insightful and strong motivation for the forthcoming study!

Thank you, we appreciate the positive feedback!

2. Data and Methods
L117: Consider indicating why ERA5 data is used on a coarser 0.5-degree grid rather than its original 0.25-degree grid.

Given that this study focuses on synoptic- and large-scale dynamics, variations on scales smaller than 500 km are not relevant to our analysis. In fact, such scales are filtered out using a spherical harmonics filter that truncates such scales. Therefore, variations that would become visible on a finer grid are filtered out by our method. We added a sentence in the manuscript in lines 120-121.

L128–130: How sensitive are subsequent results to the chosen spatial domains? For instance, the NA domain captures much more of the landmass over North America compared to the NP domain – does this potentially affect the results? Furthermore, it appears the NP domain for EKE misses the climatological maximum, whereas the NA domain is favorably located with respect to the EKE maximum. Could the authors comment on the degree to which these domain choices impact the results?

We thank the reviewer for the valid question. The rationale behind the chosen domains is that we are interested in the direct interaction between the jet and synoptic activity (in particular, the linear stage of the Rossby wave life cycle). The domains are therefore selected such that they include the climatological jet maximum, as well as the storm track entrance region where baroclinic instability is dominant (rather than barotropic wave-mean-flow interaction).

Nevertheless, we tested the sensitivity of the U-EKE relationship by repeating our analyses for two additional domain choices in both basins: (i) domains of the same size as in the manuscript but shifted 10 degrees to the east (see Fig. R1), as well as (ii) larger domains with the same western bound, extending 20 degrees further eastward (see Fig. R2). Shifting the NP domain by 10 degrees to the east allows to assess whether the larger fraction of landmass captured in the current NA domain compared to the NP domain affects the results, while increasing the size 20 degrees toward the east allows to include regions of climatologically higher EKE in the NP. Both figures reveal similar relationships to those found in our study. The correlations for the domain in the manuscript and the modified domains are listed in the following table, further confirming the robustness of our analysis.

|          | Jan (NP) | July (NP) | Jan (NA) | July (NA) |
|----------|----------|-----------|----------|-----------|
| original | -0.38    | -0.04     | -0.23    | -0.17     |
| shifted  | -0.36    | -0.01     | -0.24    | -0.16     |
| large    | -0.35    | -0.02     | -0.16    | -0.13     |

[Figure]

*Figure R1: Same as Fig. 5 in the revised manuscript but for the shifted domain.*

[Figure]

*Figure R2: Same as Fig. 5 in the revised manuscript but for the larger domain.*

**L130: How does this metric work under situations in which there may be multiple jets present at a given longitude?**

Being based on the 10-day low-pass filtered zonal wind, our metric filters short-lived jet maxima. In addition, by focusing on the western basins, we avoid capturing the (climatological) African jet. If nonetheless two persistent jets are present at a given longitude within our domain, the stronger one is considered, because we detect the wind maximum at each longitude and then average over all longitudes (within our domain). Thus, the jet strength will be dominated by the jet that extends over most of the domain but two jets at the same longitude are not separated from each other by the method. We would argue that this is a useful measure for the background jet's strength in the domain of interest.

**L139–140: Could the authors provide some more motivation regarding these choices for intensification rate and minimum sea-level pressure? For instance, grounding these values in the cyclone climatologies for each region could be a way to provide objective support for their selection.**

We thank the reviewer for this comment. We choose a 990 hPa threshold for the minimal central SLP to select only relatively strong cyclones, similar to e.g. Zhang and Colle (2017). Such a threshold truncates the left-skewed intensity distribution near its peak for both basins (see Lodise et al., 2022, and Neu et al., 2013). To ensure that only cyclones exhibiting a clear intensification phase are included, we set the (relatively weak) criterion that each track has to deepen by at least 10 hPa throughout their entire life cycle. We added explanations in the manuscript (lines 151-154).

3. The fundamental jet-storm track relationship on two timescales
L174: The distribution during March also looks less similar to the distributions during DJF for the NA, which might be worth mentioning.

Thank you for this remark, that is a very good point. We changed the formulation in the manuscript in line 190 from 'Apart from November' to 'In DJF'.

L198–206: I am having a bit of difficulty understanding the methodology for Figs. 2c,d, unfortunately. In particular, it is a bit unclear to me how the 120 timesteps are selected and averaged to produce the plots. For example, are the 120 weakest time steps selected across all years or just within each individual year? Given the varied approaches utilized within the manuscript, it might be beneficial to add either a conceptual diagram or table that summarizes the details of the different methodologies in order to help keep everything straight.

We thank the reviewer for pointing out which aspects of the methods needed to be clarified. First, the time steps are sorted using the entire time series of 40 years. Next, the weakest 120 time steps are selected and averaged, followed by averaging across the next 120 time steps in the sorted array. Figure R1 might help to better understand the method. The schematic represents an example time series. T1 to T10 represent the bins used for averaging in Fig. 2a,b, where 30 consecutive days are binned together and averaged every 10 days to produce a sliding mean time series. In contrast, Fig. 2c,d represent averages over bins of similar U containing the same number of time steps as T1-T10, in this figure U1-U4. Here we do not produce sliding means to ensure that the different means are largely independent, resulting in a smaller number of averages (4 compared to 10 in this case). We added more detail to the explanations in the manuscript (lines 213-225) and illustrated the method by adding Fig. R3 (Fig. 4 in the manuscript).

[Figure]

*Figure R3: Schematic illustrating the two averaging methods. The blue dots represent an example time series of background jet strength U shown in Fig. 2a,b.. The black and gray brackets indicate the temporal bins used to compute 30-day (120-timestep) running means. The red brackets indicate bins containing 120 timesteps grouped by similar U values, which are averaged to produce the results shown in Fig. 2c,d.*

L249: I believe the wrong section is referenced at the end of this line.

We corrected this in the manuscript.

L255–264: These explanations are really helpful in deducing how the averaging methodology works, and I think some of these details could be added earlier in section 3 to assist with the initial interpretation of results from Figs. 2 and 4.

We thank the reviewer for this suggestion on how to make the methodology more understandable to the reader. We added some of the explanations on how the averages in our new method are calculated earlier in Sect. 3 (lines 213-225).

4. The effect of jet width
L367: Are these correlations specifically calculated only for values above a certain wind speed? If so, consider specifying that here.

There is no threshold used for windspeed, it is simply calculated for DJF.

5. Implications of different jet states in DJF for eddy and cyclone characteristics
L427–434: Much of this information, save for the details of the statistical significance test, are provided in the Fig. 9 caption and likely could be omitted for brevity.

Thank you for pointing this out, we omitted parts of the information already provided in the caption.

L450: The wavier jet structure might also argue for a stronger influence from diabatic processes in the weak jet cases too. Might there be a possibility to highlight to this

effect as part of the story? Admittedly, this certainly could be an interesting avenue for future work too.

Thank you for this interesting idea. Indeed, the wavier jet structure is related to more meridional and thus ascending motion, which is likely linked to stronger diabatic processes. These in turn might amplify the downstream ridge. This is particularly interesting when considering the increasing importance of diabatic processes in a warmer climate. We included this idea in our discussion (lines 586-592)

Figures and Tables:
Fig. 2: Could the correlations referenced in the text in L156–157 be calculated and plotted on Figs. 2a,b?

The data points in Fig. 2a,b are 30-day averages computed every 10 days, such that the same timesteps contribute to multiple data points. The points are therefore highly correlated. In addition, the correlations mentioned in the text are dependent on which U-interval is observed. We would thus prefer to keep the statement qualitative. The correlations for 30-day averages without overlap are given in Tab. 1.

Fig. 5: Is there a reason why the normalized values for EKE in panel (b) do not extend to 1 for the full year, but do for the winter? Discussion of the interpretation of these normalized values being equal to 1 (or –1) might assist a reader. Also, how come the hatching in (c) extends across the colored bars? Could the statistically significant correlations be highlighted with an asterisk?

Thank you for pointing this out, we agree that some aspects of this figure require further explanation. Originally the two bars corresponding to one of the basins were normalised with respect to the larger one, but we agree with you that this is confusing. We adjusted the figure such that the bar height is normalised with respect to the all-year bar. Negative values arise because of negative correlations/covariances. In (c) the hatching extends across the coloured bars because the coloured part representing the within-month covariance is negative, thus counteracting the hatched between-months covariance. We have added explanation for these features in the manuscript (lines 316-318). All correlations are statistically significant, as we now specified in the manuscript.

Fig. 8: These results are very interesting, but would it be possible to produce plots that show the difference between the strong and weak jet terciles and evaluate those differences for statistical significance?

We added panels with the differences and significances (Fig. 9 in the revised manuscript).

Fig. 9: The difference between the violet and black stippling is a bit difficult to differentiate. Could different colors be used, perhaps?

We thank the reviewer for pointing this out. We adjusted the colour of the stippling.

Fig. B1: It is a bit confusing that different y-axes are used for terms (i) and (ii), is there a particular motivation as to why different axes are used?

The different axes were chosen because neglecting the first term in Eq. B3 leads to differences between the two expressions (i) and (ii) shown in Fig. B1. However, we agree that this can be seen as misleading and modified the figure to display the same axis for both terms in the manuscript.

**Reviewer 2**

The present paper presents a thorough statistical investigation of the relationship of transient eddy activity (or storm-track activity) with westerly jet intensity and meridional width for the North Atlantic and Pacific. By applying a novel classification method of jet intensity and meridional width to reanalysis data, the authors have revealed that the relationship between the eddy activity and time-mean flow tend to differ fundamentally between inter-monthly and sub-monthly time scales. They have also revealed that for a given jet intensity between weak and modest levels, a narrower jet tends to accompany weaker eddy activity than a broader jet. The findings in this work can be a clue for fully understanding the counterintuitive "midwinter minimum of the North Pacific storm-track activity". I consider this work to be worthy of being published, since the finding of those robust relationships makes an important contribution to the storm-track dynamics, although they still require deeper dynamical understanding. However, I find some aspects of the manuscript that require clarification and improvement as listed below. I therefore consider a minor revision is necessary to properly address those comments before the acceptance of this paper.

We thank the reviewer for the thoughtful and constructive feedback on the manuscript and their overall positive assessment of the work's significance. We greatly appreciate the insight and expertise put into the review. Below, we provide responses to the individual comments and outline how we improved the manuscript accordingly.

Major scientific comments:
[A] L100~103: In a line similar to Schemm et al. (2021), Okajima et al. (2022, J. Clim., 35 (4), 1137–1156) investigated the midwinter minimum in the North Pacific storm-track activity. They found that the net eddy conversion/generation rate normalized by the eddy total energy is indeed reduced in midwinter, to which a reduced conversion rate of baroclinic energy gain and an increase rate of barotropic energy loss in midwinter both contribute. It is a relevant work, I believe, to be cited at the end of this paragraph.

We thank the reviewer for pointing us toward this relevant work and cited it in this context (lines 78, and 87-89)

[B] L151~166: In these two paragraphs, statistical analyses by Nakamura (1992, N92) are reviewed. For deeper understanding through fairer comparison with this study, the review should include the following aspects. Specifically, N92 used eddy statistics and

mean-flow properties based on sampling not only for individual calendar months but also for 31-day moving windows, which should be mentioned properly, the latter of which is equivalent to the 30-day moving average used in this work. In fact, N92 used the moving average for constructing scatter plots between eddy statistics (activity) and a given mean flow property, including zonal wind speed or meridional temperature gradient. So, any description that may lead to such misunderstanding that N92 only used statistics for individual calendar months should be avoided. Note that Okajima et al. (2022) also used statistics for 31-day moving windows for their analyses. Another important aspect for the scatter plots in N92 is that the mean flow properties were sampled along the instantaneous local storm-track axis defined as latitudinal maxima of eddy amplitude, which differs from the present study. Therefore, the westerly wind speed and associated baroclinicity plotted in the scatter plots in N92 tend to be weaker than the corresponding values in this study especially over the midwinter North Pacific, where the storm-track tends to be located poleward of the intense narrow jet stream.

We adapted our description of the temporal averaging method in both our study and past work to avoid any misunderstandings (e.g. lines 164-167). In addition, for a clearer comparison, we specified the differences between our study and N92 concerning the region where the westerly wind speed is sampled (lines 169-171).

[C] L156: "a monthly mean, with monthly means computed around a central date every 10 d" makes no sense, and even "monthly means computed around a central date every 10 d" is quite misleading. This is because "monthly mean" typically signifies averaging over a given calendar month. Going through the manuscript, it will be finally realized that "monthly mean" or "monthly average" in this paper actually denotes "30-day running mean". If this is indeed the case, "a monthly mean, with monthly means computed around a central date every 10 d" should be replaced with "statistics based on 30-d running means evaluated at 10d intervals". In the rest of the manuscript, "monthly mean" should be replaced for clarification with "30-day moving average" or "monthly-scale averaging", wherever necessary. Individual occasions for such replacement are given as "minor comments" below.

We thank the reviewer for pointing this out. Indeed, "30-d running means" is a more precise description of the employed method compared to "monthly means". We implemented these changes in the manuscript.

[D] L236•237: "periods of enhanced baroclinic conversion are followed by a weakened U and increased EKE" sounds quite interesting, but I wonder if such a lead-lag relationship has been robustly extracted in this work through statistical analysis. The same comment applies also to a similar statement in L479~481.

We thank the reviewer for the question. We have performed an analysis examining the lead-lag variation of U and EKE around transitions of U in DJF from its upper to middle tercile. Figure R4 shows the evolution of the 2-day running mean of U and EKE before and after such transitions. This analysis is similar to that of Ambaum and Novak (2014), who investigated the evolution of heat fluxes and baroclinicity and demonstrated that baroclinicity decreases following spikes in meridional heat fluxes. However, our

analysis is centered at different kinds of events, baroclinicity is quantified by U, and instead of heat fluxes we analyse EKE. EKE increases as a consequence of ascent along the sloping isentropes reducing the mean baroclinicity and converting available potential energy to EKE. Consequently, we expect EKE to increase concurrently with a decrease in U. This is what is observed in Fig. R4 and shows that the temporal evolution is in fact as described in the manuscript. We added this figure to the supplement of the manuscript. We do however not analyse the baroclinic conversion term or APE tendencies explicitly as this is beyond the scope of our work, which is why the interpretation in the manuscript is formulated in a speculative manner.

[Figure]

*Figure R4: Evolution of the 2-day running mean of U and EKE after transitions of U from upper to middle tercile in DJF. The solid lines represent the mean of all events at each lag, while the shading represents the inter-quartile range of all events.*

[E] L315~318: It is indeed the case that "the NP jet is more tightly constrained by the descending branch of the Hadley circulation" in midwinter. In addition, the NP jet is tightly constrained also by the prominent planetary-wave trough over the Far East, as pointed out by Nakamura et al. (2002, J. Clim.) as well as Nakamura and Sampe (2002, GRL). This aspect should be noted in the text.

We thank the reviewer for this excellent insight. We added the the planetary-wave to the discussion in the manuscript and cited the mentioned studies (line 349-350) .

[F] L339: As stated here, the jet with a comparable strength tends to be broader in winter than in summer (Fig. 6), which is an important finding. From a different viewpoint, the jet with a comparable width tends to be stronger in winter than in summer, reflecting the greater equator-pole temperature difference. This is especially the case over the midwinter North Pacific, where an eddy-driven subpolar jet is merged with the subtropical jet due to the prominent planetary-wave trough over the Far East, as argued by Nakamura et al. (2004, AGU Monogr. 147, 329–345).

Thank you for this comment. We point out the viewpoint in the revised manuscript that jets with comparable width tend to be stronger in winter, which is what is frequently observed in the North Pacific during midwinter (lines 379-380).

[G] L408: As mentioned here, Fig. 9 clearly shows the tendency for transient eddies over the midwinter North Pacific to be more confined meridionally under the stronger jet than under the weaker jet. This finding is in good agreement with what is indicated in Fig. 3 of Nakamura and Sampe (2002 GRL), which should be cited here. Another important aspect indicated by Fig. 9 is that the strong jet is more zonally extended and cyclone development tends to occur closer to the jet core region if compared to the weak jet condition.

Thank you for pointing this out. We draw the connection to Nakamura and Sampe (2002), in the context of the meridional confinement of the eddies in the revised version of the manuscript (line 453). In addition, we mention the differences in the jet-relative position of the intensifying cyclones (line 463-464).

[H] L485: As commented in [D]. the midwinter North Pacific jet is constrained not only by the Hadley circulation but also by the planetary-wave trough over the Far East. The description here should be modified accordingly.

Thank you for this comment, we added this aspect to the discussion (lines 533-534).

[I] L524•525: The eddy energetics modulated under the intensified narrow jet in the midwinter North Pacific has been investigated also by Okajima et al. (2022), which may be cited here.

We thank the reviewer for pointing this out, and cited Okajima et al. (2022) in this context (line 575).

Minor comments:

L22: "narrower with smaller EKE" sounds more appropriate in context than "narrower, and therefore EKE smaller".

We changed the formulation in the abstract accordingly.

L121•122: "10-day temporal high-and low-pass filters" rather than "a 10-day temporal high-and low-pass filter".

Thank you, we adjusted the formulation as suggested.

L167: Better to describe explicitly here if this jet speed classification is based on 10-day lowpass-filtered fields or 30-day running mean fields.

We clarified this in the manuscript (line 184).

p.8, L4 Fig. 2 caption: "30-d moving averages" is better than "30-d averages".

We implemented this change as suggested.

L177: "monthly-scale averaging" is more appropriate than "monthly averaging".

We changed the formulation accordingly.

L181: "simple monthly-scale averaging" sounds more appropriate than "monthly averaging".

We adjusted this as suggested.

L201: It is certainly the case that "a distinct negative relationship emerging for averaged jet velocities exceeding 55 m/s" for the NA in Fig. 2d. This feature is, however, hinted even in Fig. 2b, which should be stated here.

Thank you for pointing this out. We mention that a negative relationship is hinted also in Fig. 2b in the revised manuscript (line 175).

L222: It should be stated here how the degrees of freedom are estimated for assessing this significance.

Thank you for this interesting remark. We estimated the effective sample size with and without a correction for a lag-1 autocorrelation. The latter was done using formula 2 from Ebisuzaki (1997). Without correction the correlations for all months in both basins are significant on a 1% level. With correction this is the case for all months in the NA, as well as all months except for July and August in the NP, the latter being significant on a 5% level. We specified the significance levels for the lag-1 corrected effective sample size in the manuscript for clarity, and stated that we correct for lag-1 autocorrelation (lines 249-251).

L225: Is this tendency "EKE generally decreases with increasing jet strength" apparent on sub-monthly timescales? If so, it should be stated explicitly here.

The tendency is in fact apparent on sub-monthly timescales, which we now state explicitly in the revised manuscript (line 253).

L233: "all calendar months" is clearer than "all months".
p.11, L3 Fig. 4 caption: "individual calendar months" is better than "individual months".

Thank you, we implemented these changes as proposed.

L253: "monthly-scale averaging" is more appropriate than "monthly averaging".

We thank the reviewer for this comment and adjusted the formulation.

L267: "the EKE decrease" is better than "the decrease".

We implemented the change as suggested.

L276~280: Here, the variance/covariance decomposition is described in referring Appendix A. Please clarify that "the variance of monthly means" is based on averages for individual (calendar) months.

We thank the reviewer for this comment and clarified this in the manuscript (lines 310-311).

L351: "narrower summer jet" rather than "more narrow summer jet

We made this adjustment as suggested.

L358: Based on Fig. 7a, the threshold jet speed over NP for EKE sensitivity to jet width seems to be "70 m/s", rather than "80 m/s" as stated here.

Thank you for noticing. We changed the description of the threshold to 70 m s$^{-1}$ (line 394).

L369: "the jet cannot be regarded simply as a fixed background state" sounds more appropriate than "the jet cannot be regarded as a fixed background state".

We adjusted the formulation accordingly.

p.20, Fig. 9: The labels along the abscissa (10E, 10W) and ordinate (10N, 10S) may be misleading. Better to replace them with (+10, –10).

We replaced the labels as proposed.

P.20, L4 Fig. 9 caption: "in thick blue contours" is better than "in blue contours".

We adjusted the caption accordingly.

L441: "a reduction of the SLP anomaly" is ambiguous. "a weakening of the SLP anomaly" is better.

We changed the formulation as suggested.

L442: Isn't it "U maximum" rather than "KE maximum"? "KE" has not been defined yet.

We thank the reviewer for the comment. We defined KE in line 489 as total kinetic energy in the revised manuscript and added the abbreviation '(KE)' in the caption of Fig. 9, so that the reader can directly relate it to the figure. We show this quantity in Fig. 10a,b rather than U, because we would like to highlight the instantaneous picture in this discussion.

What is meant by "left jet exit region"? Isn't it simply "jet exit region?

Thank you for this question. We would like to specify that it is the left exit region, as it is known as a region of ascent and cyclone intensification, which is relevant in this context.

Isn't it "U is greater by definition" rather than "KE is higher by definition"?
"wavy U pattern" rather than "wavy KE pattern"?

Thank you for this question. We would like to discuss the total wind in this context.

"when the jet is stronger and more zonal" is more specific than "when the jet is more zonal".

The strength is already mentioned in the first part of the sentence, where we refer to the 'strong jet cases' (line 502).

"monthly-scale averages" is more appropriate than "monthly averages".

We changed the formulation as suggested.

L483: "monthly-scale statistics" sounds more appropriate than "monthly averaged values".

We adjusted the formulation accordingly.

**References:**

Ambaum, M.H.P. and Novak, L.: A nonlinear oscillator describing storm track variability. Quart. J. Roy. Meteor. Soc., 140: 2680-2684, https://doi.org/10.1002/qj.2352, 2014.

Ebisuzaki, W.: A method to estimate the statistical significance of a correlation when the data are serially correlated. *J. Climate*, 10, 2147-2153, https://doi.org/10.1175/1520-0442(1997)010<2147:AMTETS>2.0.CO;2, 2017.

Lodise, J., Merrifield, S., Collins, C., Rogowski, P., Behrens, J., & Terrill, E.: Global climatology of extratropical cyclones from a new tracking approach and associated wave heights from satellite radar altimeter. *J. Geophys. Res.: Oceans*, 127, e2022JC018925, https://doi.org/10.1029/2022JC018925, 2022.

Neu, U., Akperov, M. G., Bellenbaum, N., Benestad, R., Blender, R., Caballero, R., Cocozza, A., Dacre, H. F., Feng, Y., Fraedrich, K., Grieger, J., Gulev, S., Hanley, J., Hewson, T., Inatsu, M., Keay, K., Kew, S. F., Kindem, I., Leckebusch, G. C., Liberato, M. L. R., Lionello, P., Mokhov, I. I., Pinto, J. G., Raible, C. C., Reale, M., Rudeva, I.,

Schuster, M., Simmonds, I., Sinclair, M., Sprenger, M., Tilinina, N. D., Trigo, I. F., Ulbrich, S., Ulbrich, U., Wang, X. L., & Wernli, H.: IMILAST: A community effort to intercompare extratropical cyclone detection and tracking algorithms: *Bull. Amer. Meteor. Soc.*, *94*(4), 529-547, https://doi.org/10.1175/BAMS-D-11-00154.1, 2013.

Zhang, Z., and Colle, B. A.: Changes in extratropical cyclone precipitation and associated processes during the twenty-first century over Eastern North America and the Western Atlantic using a cyclone-relative approach. *J. Climate*, 30, 8633–8656, https://doi.org/10.1175/JCLI-D-16-0906.1, 2017.

---

## Referee Report (RR1)

**"A New Look at the Jet-Storm Track Relationship in the North Pacific and North Atlantic"**

**Authors:** Zilibotti, Wernli, and Schemm

**Recommendation:** Minor Revisions

**Overview:**

Overall, I am very pleased with the efforts the authors have made to address my comments and those of the other reviewer. The manuscript, in its current form, is very well written, supported robustly by analyses, and very insightful. The authors are commended for producing a fantastic and thought-provoking study. I only have a few minor comments for the authors to consider prior to publication.

**Minor / Specific Comments:**

*1. Introduction*
L53: Consider adding a sentence that briefly summarizes these two seeding branches.

*2. Methods*
L149: If there is a pertinent figure from these studies that helps to visualize this distribution, consider referencing that here, as well.

*3. The fundamental jet-storm track relationship on two timescales*
L189–193: This result certainly appears to be qualitatively true, but it's not as obvious of a comparison as the aforementioned correlations that are discussed on L169. Could some quick statistics be calculated for the magnitude of the inter-quartile range (IQR) to further support this claim? Similar calculations could also be made when discussing IQRs in L221.

L287: I am not convinced that the black dot at U=90 m/s is exclusively from DJF since there is a pink regression line that also extends to similar values for the NP. Could the authors clarify or make a further revision to the text?

Fig. 6: The y-axis for this figure is a bit confusing since it may inadvertently imply that the plot shows the variance divided by the covariance. Consider an alternative way of expressing this label to eliminate potential confusion.

L299: It is a bit unconventional that these traits are listed out of order compared to how they are shown in Fig. 6. Consider an edit to the text that lists the variables in the same order as they are shown in Fig. 6.

L351: Consider referencing Fig. 2 to help remind the reader where to verify this prior result.

*5. Implications of different jet states in DJF for eddy and cyclone characteristics*
L436: Consider breaking this paragraph into two separate paragraphs, with the second paragraph beginning with the discussion of the eddy orientation.

*6. Summary and conclusions*

L531–532: The last half of this sentence seems a bit out of place (i.e., the part that begins with "can be even better understood…). Consider a revision that further clarifies the discussion. Perhaps a solution could be to swap "as well as" with "or"?

Section wide: A lot of references are made to results from prior figures as part of this synthesizing discussion. Consider referencing the pertinent figures that support various claims with parenthetical references to help orient a reader who may start by reading the conclusion section as an "executive summary" of the work.

---

## Author Response (AR2)

**Reply document for egusphere-2025-3605**

**A new look at the jet-storm track relationship in the North Pacific and North Atlantic**

by Nora Zilibotti, Heini Wernli, and Sebastian Schemm

We thank reviewer 1 for the positive feedback following our revision and for the additional minor comments. This document presents the reviewers' comments in blue and our responses and changes in black. The line numbers correspond to the lines in the marked-up manuscript version showing the changes.

**Reviewer 1**

Overall, I am very pleased with the efforts the authors have made to address my comments and those of the other reviewer. The manuscript, in its current form, is very well written, supported robustly by analyses, and very insightful. The authors are commended for producing a fantastic and thought-provoking study. I only have a few minor comments for the authors to consider prior to publication.

*1. Introduction*
L53: Consider adding a sentence that briefly summarizes these two seeding branches.

We added a sentence describing the two branches (L53-54).

*2. Methods*
L149: If there is a pertinent figure from these studies that helps to visualize this distribution, consider referencing that here, as well.

We added the reference to a specific figure from one of the studies (L150).

*3. The fundamental jet-storm track relationship on two timescales*
L189–193: This result certainly appears to be qualitatively true, but it's not as obvious of a comparison as the aforementioned correlations that are discussed on L169. Could some quick statistics be calculated for the magnitude of the inter-quartile range (IQR) to further support this claim? Similar calculations could also be made when discussing IQRs in L221.

The first part of the comment addresses the statement that the NA jet exhibits strong variability on sub-monthly timescales during extended winter, such that averaging over 30-day windows effectively averages over a wide range of jet velocities. This should be interpreted as a relative statement: the range of jet strengths sampled within individual months is comparable to the total range of jet strengths observed over the entire extended winter. To demonstrate this, we compute the IQR of jet velocities separately for each extended winter month of each year. We then take the median of these IQRs (to quantify a representative IQR of jet velocities within an individual month) and compare it to the IQR of jet velocities in the full extended winters over the entire 43-year

period. For the NA, this ratio is 0.76, indicating that a typical month captures a large part of the overall jet strength variability in extended winter, compared to 0.61 in the NP. These statistics are in line with the statement, qualitatively drawn from Fig. 3, and further quantified in Fig. 6, that the extended winter jet variability in the NA is more strongly dominated by the sub-monthly variability than in the NP.

In L221 (in the previous version of the manuscript) we state that although the IQR of the bins seems (at least visually) to be reduced with the new binning method, compared to the temporal bins, the IQRs remain large. However, a calculation of the median bin IQR for the two methods does not show a significant difference. This is not entirely surprising, as for the temporal binning method, even though we do not capture the direct relation between U and EKE as well, the temporal correlation of EKE reduces the variability within bins.

We therefore removed the part of the sentence that compares the IQR, only keeping the statement that the IQR remains large with the new method (L225-226). We thank the reviewer for helping us find this imprecision.

L287: I am not convinced that the black dot at U=90 m/s is exclusively from DJF since there is a pink regression line that also extends to similar values for the NP. Could the authors clarify or make a further revision to the text?

We thank the reviewer for this comment. We modified the text to include March in the contributing months for the black dot U=90m/s.

Fig. 6: The y-axis for this figure is a bit confusing since it may inadvertently imply that the plot shows the variance divided by the covariance. Consider an alternative way of expressing this label to eliminate potential confusion

We modified the y-axis to 'normalised var and cov' to make it less confusing for the reader.

L299: It is a bit unconventional that these traits are listed out of order compared to how they are shown in Fig. 6. Consider an edit to the text that lists the variables in the same order as they are shown in Fig. 6.

We modified the order to correspond to the panels from Fig. 6.

L351: Consider referencing Fig. 2 to help remind the reader where to verify this prior result.

The content of this sentence corresponds to the results from Fig. 5. We added a reference to help remind the reader where to verify this result, as suggested.

5. *Implications of different jet states in DJF for eddy and cyclone characteristics*
L436: Consider breaking this paragraph into two separate paragraphs, with the second paragraph beginning with the discussion of the eddy orientation.

We have broken the paragraph into two as suggested in the revised version of the manuscript.

*6. Summary and conclusions*
L531–532: The last half of this sentence seems a bit out of place (i.e., the part that begins with "can be even better understood…). Consider a revision that further clarifies the discussion. Perhaps a solution could be to swap "as well as" with "or"?

Thank you for pointing out this confusing sentence. We simplified the wording in the revised document (L534-535).

Section wide: A lot of references are made to results from prior figures as part of this synthesizing discussion. Consider referencing the pertinent figures that support various claims with parenthetical references to help orient a reader who may start by reading the conclusion section as an "executive summary" of the work.

We added references to the pertinent figures throughout this section.